# GENERALIZATION BOUNDS FOR MAGNITUDE-BASED PRUNING VIA SPARSE MATRIX SKETCHING

## ABSTRACT

Magnitude-based pruning is a popular technique for improving the efficiency of Machine Learning, but also surprisingly maintains strong generalization behavior. Explaining this generalization is difficult, and existing analyses connecting sparsity to generalization rely on more structured compression than simple magnitude-based weight dropping. We circumvent the need for structured compression by using recent random matrix theory and sparse matrix sketching results to more tightly tie the connection between pruning-based sparsity and generalization and provide bounds on how Magnitude-Based Pruning and Iterative Magnitude Pruning affects generalization. We empirically verify that our bounds capture the connection between pruning-based sparsity and generalization more than existing bounds.

## 1 INTRODUCTION

Overparameterized neural networks are often used in practice as they achieve remarkable generalization errors (Goodfellow et al., 2016). However, their immense size makes them slow and expensive to run during inference. Machine learning (ML) practitioners often employ *Magnitude-Based pruning* (MBP) to amend this computational complexity (Han et al., 2015). After training large neural networks, the parameters or matrix elements within the model with the smallest magnitude are set to $0$. This significantly reduces the memory requirements of the model and the inference time. However, MBP has also been shown to induce little generalization error and, in fact, often reduces the generalization error compared to the original model (Han et al., 2015; Li et al., 2016; Cheng et al., 2017). Examining where and why strong generalization happens can help build learning algorithms and models that generalize better (Foret et al., 2020; Le et al., 2018).

However, theoretical analyses of why MBP achieves strong generalization errors still need to be made. Existing analyses connecting compression or sparsity to generalization often assume more structured sparsity, such as coresets (Baykal et al., 2019) or weight-norm reduction Neyshabur et al. (2017). Providing such analyses is challenging for several reasons. First, removing the smallest weights is a relatively unstudied operation in linear algebra, and only a few tools are available to analyze the properties of pruned matrices. Second, it is difficult to characterize the distribution of weights after training and pruning.

However, Arora et al. (2018) provided a tool that more comprehensively analyzes the generalization error of models with fewer parameters effectively. Specifically, they upper bounded the generalization error of a large neural network when compressed. Their bound is split into two parts: the amount of error introduced by the model via the compression and the number of different parameters in the compressed model. We use this primary tool to show that the outputs from MBP generalize well since they do not introduce much error and have fewer parameters. We prove both of these phenomena with a few simple assumptions. Namely, given some justifiable assumptions on the distribution of the trained weight parameters, we develop an upper bound of the error the pruned neural network suffers with high probability. Moreover, using counting techniques or matrix sketching techniques yields efficient methods to count the number of efficient parameters in pruned weight matrices. Both simple counting techniques and sparse matrix sketching techniques achieve tight bounds and we present both styles of analysis in this paper.

Combining the two parts of the bound, we get a novel generalization error bound that is competitive with state-of-the-art generalization bounds. Moreover, to our knowledge, this is the *first* analysis that

connects pruning-based sparsity to generalization. On heavily pruned models, our bounds can capture the sparsity of the models to yield tighter bounds than existing modern bounds empirically. We empirically verify the success of our novel approach on the MNIST and CIFAR10 datasets where our bound is several orders of magnitude better (at least $10^7$ times better on CIFAR10, refer Figure 3b) than well-known standard bounds. We extend our framework to show that using Iterative Magnitude Pruning (IMP) or Lottery Tickets (Frankle & Carbin, 2018) also generalizes. Namely, Malach et al. (2020) shows that IMP produces results with small error and few nonzero parameters. We use matrix sketching to efficiently count the number of parameters in a usable way for our generalization bound. This results in a strong generalization bound that, to our knowledge, has only been analyzed empirically (Bartoldson et al., 2020; Jin et al., 2022). While analyses detailing pruning error bounds and parameter counting for pruned networks have been done before, they have not been utilized to deliver generalization bounds for pruned matrices. Moreover, we emphasize that while generalization bounds exist for **sparse models**, the arguments in these papers assume structure to the sparsity not existing in pruned models. We highlight that our analyses can extend to other forms of pruning that have low error and have sparsity evenly distributed throughout the matrix. For example, we extend our analysis to prove the generalization bounds of Lottery Tickets.

**Contributions** We circumvent the need for structured compression in traditional generalization bounds by using matrix sketching to yield one of the first connections between pruning-based sparsity and generalization. We begin by provably and empirically verifying the well-observed phenomenon that pruning does not induce large errors in models. We then provably and empirically demonstrate that the sparsity induced by pruning is distributed roughly evenly across all rows and columns. We then use sparse matrix sketching or simple counting arguments to demonstrate that pruned matrices can be represented with relatively fewer effective parameters. Combining these insights with compression bounds yields generalization bounds that can exploit the sparsity from pruning to yield tighter bounds than the generalization literature. We also empirically verify that our bounds are tighter than existing generalization bounds on pruned MLPs on CIFAR10 and MNIST. Using our proof framework, we extend the analysis to prove the generalization properties of Iterative Magnitude Pruning.

## 2 RELATED WORKS

### 2.1 NORM-BASED GENERALIZATION BOUNDS

In recent years, many works have studied how to use parameter counting and weight norms to form tighter generalization bounds as an evolution from classical Rademacher Complexity and VC dimension. Galanti et al. (2023) uses Rademacher Complexity to develop a generalization bound for naturally sparse networks such as those from sparse regularization. Neyshabur et al. (2015) studies a general class of norm-based bounds for neural networks. Moreover, Bartlett & Mendelson (2002) used Rademacher and Gaussian Complexity to form generalization bounds. Long & Sedghi (2020) gives generalization error bounds for Convolutional Neural Networks (CNNs) using the distance from initial weights and the number of parameters that are independent of the dimension of the feature map and the number of pixels in the input. Daniely & Granot (2019) uses approximate description length as an intuitive form for parameter counting. We note that similar parameter counting techniques for Lottery Tickets are present in Burkholz (2022); Pensia et al. (2020).

### 2.2 PRUNING TECHNIQUES

While MBP is one of the most common forms of pruning in practice, other forms exist. Collins & Kohli (2014) induce sparsity into their CNNs by using $\ell_1$ regularization in their training. Molchanov et al. (2017) develops iterative pruning frameworks for compressing deep CNNs using greedy criteria-based pruning based on the Taylor expansion and fine-tuning by backpropagation. Liu et al. (2017) use Filter Sparsity alongside Network Slimming to enable speedups in their CNNs. Ullrich et al. (2017) coins soft-weight sharing as a methodology of inducing sparsity into their bounds. Moreover, Hooker et al. (2019) empirically studied which samples of data-pruned models will significantly differ from the original models. Many works use less common pruning methods such as coresets (Mussay et al., 2019) or the phenomenon of IMP (Frankle & Carbin, 2018; Malach et al., 2020).

## 3 PRELIMINARY

### 3.1 NOTATION

We consider a standard multiclass classification problem where for a given sample $x$, we predict the class $y$, which is an integer between 1 and $k$. We assume that our model uses a learning algorithm that generates a set of $L$ matrices $\mathbf{M} = \{\mathbf{A}_1, \ldots, \mathbf{A}_L\}$ where $\mathbf{A}_i \in \mathbb{R}^{d_1^i \times d_2^i}$. Here, $d_1^i, d_2^i$ are the dimensions of the $i$th layer. Therefore, given some input $x$, the output of our model denoted as $\mathbf{M}(x)$ is defined as $\mathbf{M}(x) = \mathbf{A}_L \phi_{L-1}(\mathbf{A}_{L-1}\phi_{L-2}(\ldots \mathbf{A}_2\phi_1(\mathbf{A}_1 x)))$, thereby mapping $x$ to $\mathbf{M}(x) \in \mathbb{R}^k$. Here, $\phi_i$ is the activation function for the $i$th layer of $L_i$ Lipschitz-Smoothness. When not vague, we will use the notation $x^0 = x$ and $x^1 = \mathbf{A}_1 x$ and $x^2 = \mathbf{A}_2\phi_1(\mathbf{A}_1 x)$ and so on. Given any data distribution $\mathcal{D}$ the expected margin loss for some margin $\gamma > 0$ is defined as

$$R_\gamma(\mathbf{M}) = \mathbb{P}_{(x,y)\sim\mathcal{D}}\left[\mathbf{M}(x)[y] \leq \gamma + \max_{j \neq y}\mathbf{M}(x)[j]\right].$$

The population risk $R(\mathbf{M})$ is obtained as a special case of $R_\gamma(\mathbf{M})$ by setting $\gamma = 0$. The empirical margin loss for a classifier is defined as $\hat{R}_\gamma(\mathbf{M}) = \frac{1}{|\mathcal{S}|}\sum_{(x,y)\in\mathcal{S}}\mathbb{I}\left(\mathbf{M}(x)[y] - \max_{j \neq y}(\mathbf{M}(x)[j]) \leq \gamma\right)$, for some margin $\gamma > 0$ where $\mathcal{S}$ is the dataset provided (when $\gamma = 0$, this becomes the classification loss). Intuitively, $\hat{R}_\gamma(\mathbf{M})$ denotes the number of elements the classifier $\mathbf{M}$ predicts the correct $y$ with a margin greater than or equal to $\gamma$. Moreover, we define the size of $\mathcal{S}$ to be $|\mathcal{S}| = n$. We will denote $\hat{\mathbf{M}} = \{\hat{\mathbf{A}}^1, \ldots, \hat{\mathbf{A}}^L\}$ as the compressed model obtained after pruning $\mathbf{M}$. The generalization error of the pruned model is then $R_0(\hat{\mathbf{M}})$. Moreover, we will define the difference matrix at layer $l$ as $\Delta^l = \mathbf{A}^l - \hat{\mathbf{A}}^l$. Now that we have formally defined our notation, we will briefly overview the main generalization tool throughout this paper.

### 3.2 COMPRESSION BOUNDS

As compression bounds are one of the main theoretical tools used throughout this paper, we will briefly overview the bounds presented in Arora et al. (2018). Given that we feed a model $f$ into a compression algorithm, the set of possible outputs is a set of models $G_\mathcal{A}$ where $\mathcal{A}$ is a set of possible parameter configurations. We will call $g_A$ as one such model corresponding to parameter configuration $A \in \mathcal{A}$. We define compressibility explicitly in the following definition.

**Definition 3.1.** *If $f$ is a classifier and $G_\mathcal{A} = \{g_A | A \in \mathcal{A}\}$ be a class of classifiers with a set of trainable parameter configurations $\mathcal{A}$ and fixed string $s$. We say that $f$ is $(\gamma)$-compressible via $G_\mathcal{A}$ if there exists $A \in \mathcal{A}$ such that for any $x \in \mathcal{S}$, we have for all $y$, $|f(x)[y] - g_A(x)[y]| \leq \gamma$.*

Proving generalization for a compressified classifier yields a bound that depends on both the margin and the number of parameters in the pruned model, as in the following theorem.

**Theorem 3.1.** *(Arora et al., 2018) Suppose $G_{\mathcal{A},s} = \{g_{A,s} | A \in \mathcal{A}\}$ where $A$ is a set of $q$ parameters each of which can have at most $r$ discrete values and $s$ is a helper string. Let $\mathcal{S}$ be a training set with $n$ samples. If the trained classifier $f$ is $(\gamma, \mathcal{S})$-compressible via $G_\mathcal{A}$, then there exists $A \in \mathcal{A}$ with high probability over the training set , $R_0(g_A) \leq \hat{R}_\gamma(f) + \mathcal{O}\left(\sqrt{\frac{q\log r}{n}}\right).$*

It is to be noted that the above theorem provides a generalization bound for the compressed classifier $g_A$, not for the trained classifier $f$. Therefore, the two parts of forming tighter generalization bounds for a given compression algorithm involve bounding the error introduced by the compression, the $\gamma$ in $\hat{R}_\gamma(f)$, and the number of parameters $q$ after compression. We demonstrate that we can achieve both with traditional MBP.

### 3.3 PRELIMINARY ASSUMPTIONS

Analyzing the effects of pruning is difficult without first understanding from which distribution the weights of a trained model lie. This is a complex and open question in general. However, Han et al.

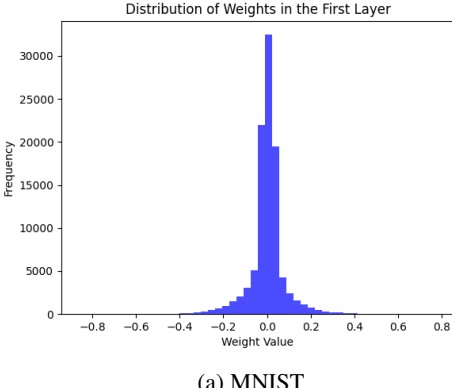

(a) MNIST

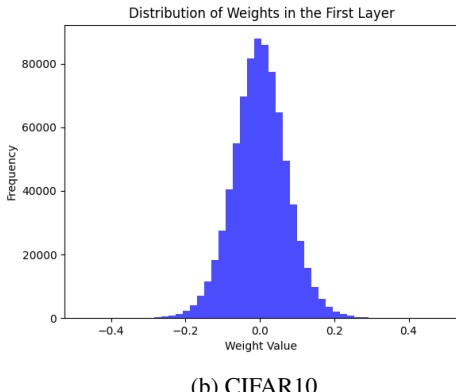

(b) CIFAR10

Figure 1: Here, we plot the empirical distribution over the weights to verify our Guassian Assumption

(2015) made the empirical observation that weights often lie in a zero-mean Gaussian distribution, such as in Figure 7 in Han et al. (2015). We will assume this to be true, that the distribution of weights follows a normal distribution with $0$ mean and variance $\Psi$. Here, we state the main preliminary assumptions that we will use later.

**Assumption 3.1.** *For any $l \in [L], i, j \in [d_1^l] \times [d_2^l]$, $\mathbf{A}_{i,j}^l \sim \mathcal{N}(0, \Psi)$ and are independent of each other.*

This assumption states that each atom within a matrix of the learned model obeys roughly a Gaussian distribution centered at $0$ with variance $\Psi$. While a strong assumption, this is common. In fact, Qian & Klabjan (2021) assumes that the weights follow a uniform distribution to analyze the weights of pruned models. We assume a Gaussian distribution since this is more reasonable than the uniform distribution assumption. For all claims based on this assumption, we will provide empirical verification of the claim to support our analysis. We can now present the MBP algorithm we will analyze throughout this paper. We have added empirical verification of this assumption in Figure 1.

## 4 MAGNITUDE-BASED PRUNING ALGORITHM

While many versions of MBP algorithms exist, they are all based on the general framework of removing weights of small magnitude to reduce the number of parameters while ensuring the pruned model does not differ vastly from the original. We present a formulation of MBP [1] that mimics the random MBP seen often in works like Han et al. (2015); Qian & Klabjan (2021). We formally present our algorithm in Algorithm 1 below.

**Remark 4.1.** *We do not prune the diagonal elements in Algorithm 1. While not standard, this enables the use of Matrix Sketching later on for better generalization bounds. However, in Dasarathy et al. (2013), they note the necessity for the diagonal elements (i.e. elements $\mathbf{A}_{i,j}^l$ where $i = j$) being nonzero is for ease of presentation of the proof, and Matrix Sketching should still be possible with pruning the diagonal elements.*

Here, $d$ is a hyperparameter helpful for adjusting the strength of the compression. To use the

---

**Algorithm 1:** MBP

**Data:** $\{\mathbf{A}^1, \dots, \mathbf{A}^L\}, d$
**Result:** $\{\hat{\mathbf{A}}^1, \dots, \hat{\mathbf{A}}^L\}$
**for** $l \in [L]$ **do**
    **for** $i, j \in [d_1^l] \times [d_2^l]$ *and* $i \neq j$ **do**
        $X := \text{Bernoulli}\left( \exp\left( \frac{-[\mathbf{A}_{i,j}^l]^2}{d\Psi} \right) \right)$
        $\hat{\mathbf{A}}_{i,j}^l := 0$ if $X = 1$ else $\mathbf{A}_{i,j}^l$
    **end**
**end**

---

[1]While the term inside the Bernoulli random variable used as an indicator for pruning is slightly different as compared to previous literature, this is a small change that allows us to move away from the uniform distribution assumption from Qian & Klabjan (2021) to a more favorable Gaussian assumption.

generalization bounds from Section 3.2, we need to show that Algorithm 1 creates a pruned model $\hat{\mathbf{M}}$ that produces outputs similar to the original model $\mathbf{M}$. We prove this in the sections below.

## 4.1 ERROR PROOF

We begin by bounding the difference between the outputs of corresponding layers in the pruned and original models to prove that the expected difference between the pruned and original models is small. The normality assumption from Assumption 3.1 makes this much more tractable to compute. Indeed, each atom of the difference matrix $\Delta^l = \hat{\mathbf{A}}^l - \mathbf{A}^l$ is an independent and identical random variable. Bounding the $\ell_2$ norm of such a matrix relies only on the rich literature studying the norms of random matrices. In fact, from Latala (2005), we only need a bounded second and fourth moment of the distribution of each atom. To utilize this bound, we only need to demonstrate that the difference matrix $\Delta^l$ and the pruned model obtained using the compression scheme Algorithm 1 have atoms whose moments are bounded and have zero-mean, which is automatic from Assumption 3.1. Then, given a bound on the norm of $\Delta^l$, we can prove the error bound for our entire sparse network by performing an induction on an upper bound of the error throughout each layer. We can now present our error bound for our entire sparse network.

**Lemma 4.1.** *The difference between outputs of the pruned model $\hat{x}^L$ and the original model $x^L$ on any input $x$ is bounded by, with probability at least $1 - \sum_l^L \epsilon_l^{-1}$,*

$$\|\hat{x}^L - x^L\|_2 \le e\|x\|_2 \left( \prod_{l=1}^L L_l \|\mathbf{A}^l\|_2 \right) \sum_{l=1}^L \frac{\epsilon_l \Gamma_l}{\|\mathbf{A}^l\|_2}.$$

*Here,* $\Gamma_l = C \left[ \left( \sqrt{\frac{d^{\frac{3}{2}} \Psi}{(d+2)^{\frac{3}{2}}}} \right) \left( \sqrt{d_1^l} + \sqrt{d_2^l} \right) + \left( \frac{3 d_1^l d_2^l d^{\frac{5}{2}} \Psi^2}{(d+2)^{\frac{5}{2}}} \right)^{\frac{1}{4}} \right].$

We can form tighter bounds by considering what the expected maximum of $(\hat{\mathbf{A}}^l - \mathbf{A}^l)x$ is with high probability. If $d_2^l < d_1^l$, we observe that the matrix $\hat{\mathbf{A}}^l - \mathbf{A}^l$ has at most $d_2^l$ nonzero singular values. For more details, please see Appendix D. However, more than this error bound is needed to prove strong generalization bounds. We require the number of possible models after training and compression to be finite to use compression bounds. Therefore, we need to apply discretization to our compressed model to ensure that the number of models is finite.

**Empirical Verification** This analysis was based on a strong assumption that the weights are Gaussian. However, the claim made in Lemma C.5 is also empirically visible. As the pruning parameter $d$ increases, we plot how the true error induced by pruning looks compared to our predicted upper bound in Lemma C.5. We do this for MLPs on both CIFAR10 and MNIST data. We see the results in Figure 2. As empirically shown, despite using Assumption 3.1, Lemma C.5 still holds in practice since the generated error bound is larger than the true error induced by pruning. For more experimental details, see Section 7.

## 4.2 DISCRETIZATION

We now show that the prediction error between a discretization of the pruned model and the original model is also bounded. Our discretization method is simply rounding each value in layer $l$ to the nearest multiple of $\rho_l$. We will call the discretized pruned model $\tilde{\mathbf{M}}$ where the $l$th layer will be denoted as $\tilde{\mathbf{A}}^l$. We provide the following lemma bounding the norm of the difference of the layers between the pruned and the discretized model. Using this intuition, we can prove that the error induced by the discretization is small.

**Lemma 4.2.** *The norm of the difference between the pruned layer and the discretized layer is upper-bounded as $\|\tilde{\mathbf{A}}^l - \hat{\mathbf{A}}^l\|_2 \le \rho_l J_l$ where $J_l$ is the number of nonzero parameters in $\hat{\mathbf{A}}^l$ ($J_l$ is used for brevity here and will be analyzed later). Denote $x^L$ as the output of the original model and $\tilde{x}^L$ as the output of the pruned and discretized model. With probability at least $1 - \sum_{l=1}^L \epsilon_l^{-1}$, given that the parameter $\rho_l$ for each layer is chosen such that $\rho_l \le \frac{\frac{1}{L}\|\mathbf{A}^l\|_2 - \epsilon_l \Gamma_l}{J_l}$, we have that the error*

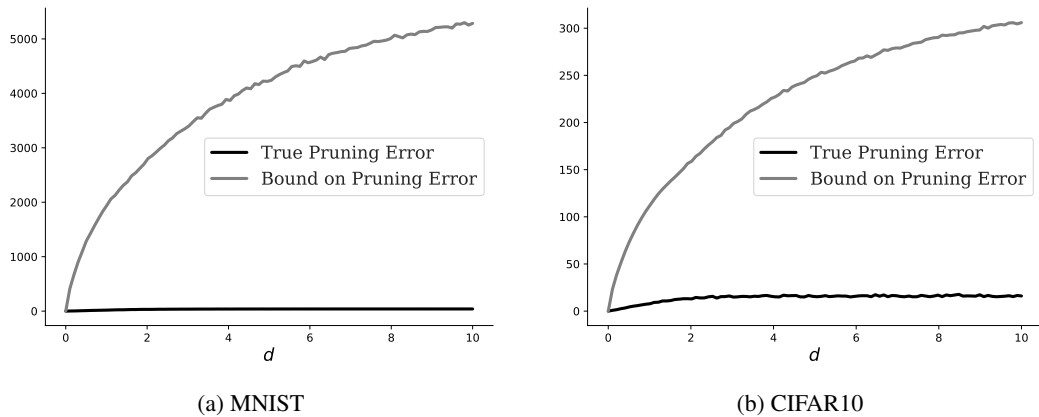

(a) MNIST                    (b) CIFAR10

Figure 2: Here, we plot the error induced by pruning and see that our error bound in Lemma C.5 holds empirically as well despite Assumption 3.1.

*induced by both discretization and the pruning is bounded by*

$$\|x^L - \tilde{x}^L\|_2 \le e\|x\|_2 \left(\prod_{l=1}^{L} L_l \|\mathbf{A}^l\|_2\right) \sum_{l=1}^{L} \frac{\epsilon_l \Gamma_l + \rho_l J_l}{\|\mathbf{A}^l\|_2}.$$

Now, we have a sufficient error bound on our MBP algorithm. Thus, as the next step, we focus on bounding the number of parameters our compressed model will have. To do this, we introduce our next significant theoretical tool: *Matrix Sketching*.

## 5    SKETCHING SPARSE MATRICES

As seen in Theorem 3.1, the generalization error depends strongly on the number of parameters. We try to count the number of possible parameterizations of the pruned model $\hat{\mathbf{M}}$ achievable by combining the learning algorithm and the compression algorithm. In the appendix, we discuss another approach by counting the number of possible sparse matrices generated by the combination of a learning algorithm and Algorithm 1. We now introduce the preliminaries and motivate the need for matrix sketching.

### 5.1    PRELIMINARY ON MATRIX SKETCHING

Here we introduce the preliminary concepts of matrix sketching. Namely, we can represent a sparse matrix $X \in \mathbb{R}^{p_1 \times p_2}$ as $Y \in \mathbb{R}^{m \times m}$ where $p_1, p_2 \ge m$. The idea of matrix sketching is to create an embedding for this matrix as $Y = AXB^\top$. Here, the matrices $A \in \{0,1\}^{m \times p_1}$, $B \in \{0,1\}^{m \times p_2}$ are chosen before the sketching is to be done. To recover the original matrix, we solve the minimization problem

$$\min_{\tilde{X} \in \mathbb{R}^{p_1 \times p_2}} \|\tilde{X}\|_1 \text{ s.t. } Y = A\tilde{X}B^\top. \tag{1}$$

If the problem from Equation (1) enjoys a unique minimum and that unique minimum is the true $X$, we can say that this sketching scheme is lossless. In such a case, all the information in $X$ is encoded in $Y$. Given such a mapping, we can use this one-to-one mapping between $Y$ and $X$ to count the number of parametrizations of $X$ using $Y$, which is of a smaller dimension. We use properties from this literature to help develop and prove the improved generalization error bounds.

We claim with matrix sketching that we can represent the space of large sparse matrices of dimension $p$ with the set of small dense matrices of dimension $\sqrt{jp} \log p$ where $j$ is the maximum number of nonzero elements in any row or column. Counting the number of parameters in small dense matrices is more efficient in terms of parameters than counting the number of large sparse matrices, thus providing a way of evasion of the combinatorial explosion. We formalize this in the following section.

## 5.2 SPARSE CASE

To apply the matrix sketching literature to our sparse matrices, we need to prove several properties of the matrices obtained using our compression scheme Algorithm 1. We introduce one such structure called $j_r, j_c$-*distributed-sparsity*, which ensures sketching can be applied to matrices. Intuitively, such a property ensures that any row or column of our sparse matrix does not contain too many nonzero elements. We formally define such intuition here.

**Definition 5.1.** *A matrix is $j_r, j_c$-distributed sparse if at most $j_r$ elements in any column are nonzero, $j_c$ elements in any row are nonzero, and the diagonal elements are all nonzero.*

The other main knowledge is how to form $A, B$ for sparse matrix sketching. If the reader is interested, we discuss how to form $A$ and $B$ alongside some intuition behind matrix sketching in Appendix F.1.

## 5.3 BOUNDS FOR SPARSE MATRIX SKETCHING

From Dasarathy et al. (2013), it can be proved that sketching the set of $j_r, j_c$-distributed sparse matrices requires only small $m$. Given a choice of $m$ and probability term $\delta$, one can show that the solution to Equation (1) matches the uncompressed value with high probability. This is mainly shown by first demonstrating that a solution exists and that the best solution to $A^{-1}YB^{-1} = \tilde{X}$ is the only solution that minimizes the $\ell_1$ norm with high probability.

**Theorem 5.1.** *(From Theorem 1 of Dasarathy et al. (2013)) Let $p = \max(d_1^l, d_2^l)$. Suppose that $A \in \{0,1\}^{m \times d_1^l}$, $B \in \{0,1\}^{m \times d_2^l}$ are drawn independently and uniformly from the $\delta$-random bipartite ensemble. Then, as long as $m = \mathcal{O}(\sqrt{\max(j_c d_1^l, j_r d_2^l)} \log(p))$ and $\delta = \mathcal{O}(\log(p))$, there exists a $c \geq 0$ such that for any given $j_r, j_c$-distributed sparse matrix $X$, sketches $AXB$ into $\tilde{X}$ results in a unique sketch for each $X$. This statement holds with probability $1 - p^{-c}$.*

**Remark 5.1.** *The theorem statement for Theorem 5.1 states that $c \geq 0$. However, in the proof, they demonstrate the stronger claim that $c \geq 2$. Therefore, the probability that Theorem 5.1 holds is at least $1 - p^{-2}$. As $p$ grows, the probability that this theorem holds approaches $1$.*

## 5.4 GENERALIZATION ERROR FROM SKETCHING

To use the above theoretical tools of matrix sketching, we must show that outputs from our compression algorithm Algorithm 1 satisfy the definitions of $j_r, j_c$-distributed-sparsity. Such a claim is intuitive and similar properties have been shown for random matrices following different distributions. Given that our trained matrices satisfy the Gaussian distribution, one row or column is unlikely to contain many nonzero elements. Here, we prove in the following lemma that the pruned model using Algorithm 1 satisfies the condition of distributed sparsity using Assumption 3.1.

**Lemma 5.1.** *With probability at least $1 - \frac{1}{\lambda_l} - (d_1^l)^{-\frac{1}{3}} - (d_2^l)^{-\frac{1}{3}}$, we have that the outputs from Algorithm 1 are $j_r, j_c$-sparsely distributed where $\max(j_r, j_c) \leq 3\lambda_l \max(d_1^l, d_2^l)\chi$ and $\lambda_l \in \mathbb{R}$. Here, $\chi = \frac{\sqrt{d+2} - \sqrt{d}}{\sqrt{d+2}}$.*

Given the above quantification of the space of sparse matrices and the bound of the error of our model, we can apply the compression bound from Arora et al. (2018). Such a compression bound intuitively depends mainly on these two values.

**Theorem 5.2.** *For every matrix $\hat{\mathbf{A}}^l$, define $j_l$ to be the $\max(j_r, j_c)$ where $j_r$ and $j_c$ are the distribution-sparsity coefficients for $\hat{\mathbf{A}}^l$. Moreover, for every matrix $\hat{\mathbf{A}}^l$, define $p_l = \max(d_1^l, d_2^l)$. Then, we have that*

$$R_0(g_A) \leq \hat{R}_\gamma(f) + \mathcal{O}\left(\sqrt{\frac{\sum_l 3\lambda_l \chi d_2^l d_1^l \log^2(p_l) \log(\frac{1}{\rho_l})}{n}}\right).$$

*This holds when $d$ is chosen such that $\gamma \geq e\|x\|_2 \left(\prod_{l=1}^d L_l \|\mathbf{A}^l\|_2\right) \sum_{l=1}^L \frac{\epsilon_l \Gamma_l + \rho_l J_l}{\|\mathbf{A}^l\|_2}$ where $J_l \leq \mathcal{O}\left(\chi d_2^l d_1^l\right)$. This claim holds with probability at least $1 - \left[\sum_{l=1}^L \lambda_l^{-1} + \epsilon_l^{-1} + p_l^{-c}\right]$.*

Here, $\chi$ depends on our hyperparameter $d$.

## 6 Generalization of Lottery Tickets

Such a generalization error proof framework for pruning applies to more than just Magnitude-based pruning. An exciting pruning approach is simply creating a very large model $G$ such that some smaller target model $\mathbf{M}$ is hidden inside $G$ and can be found by pruning. This lottery ticket formulation for pruning has seen many empirical benefits. Formally, we will call our lottery ticket within $G$ a weight-subnetwork $\widetilde{G}$ of $G$. This $\widetilde{G}$ is a pruned version of the original $G$. In fact, Malach et al. (2020) shows that for a sufficiently large $G$, there exists with high probability a pruning $\tilde{G}$ such that $\tilde{G}$ and the target function $\mathbf{M}$ differ by at most $\epsilon$. Moreover, this $\tilde{G}$ will have approximately the same number of nonzero parameters as the original target network $\mathbf{M}$. Given this, we can apply our framework, and we get the following generalization bound.

**Theorem 6.1.** *Fix some $\epsilon, \delta \in (0, 1)$. Let $\mathbf{M}$ be some target network of depth $L$ such that for every $i \in [L]$ we have $\|\mathbf{A}^i\|_2 \leq 1, \|\mathbf{A}^i\|_{\max} \leq \frac{1}{\sqrt{d_{1,i}}}$. Furthermore, let $n_{\mathbf{M}}$ be the maximum hidden dimension of $\mathbf{M}$. Let $G$ be a network where each of the hidden dimensions is upper bounded by $\mathrm{poly}\left(d_{1,0}, n_{\mathbf{M}}, L, \frac{1}{\epsilon}, \log \frac{1}{\delta}\right) := D_G$ and depth $2L$, where we initialize $\mathbf{A}^i$ from the uniform distribution $U([-1, 1])$. Moreover, with probability at least $1 - \delta - LD_G^{-c}$, there exists a weight-subnetwork $\tilde{G}$ of $G$ that obeys the generalization error of*

$$
R_0(\tilde{G}) \leq \hat{R}_{\epsilon+\epsilon_\rho}(\tilde{G}) + \mathcal{O}\left(\sqrt{\frac{[n_{\mathbf{M}} d_{0,1} \log(D_G)^2 + L n_{\mathbf{M}}^2 \log(D_G)^2] \log\left(\frac{1}{\rho}\right)}{n}}\right).
$$

*Here, $\epsilon_\rho$ is the small error introduced by discretization.*

One interesting thing to note is that this bound is only a small factor of $\log(D_G)$ worse than if we had applied a compression bound to a model of the size of the target function $\mathbf{M}$. To our knowledge, this is one of the first generalization analyses for lottery tickets.

## 7 Empirical Analysis

We study the generalization bound obtained using Algorithm 1 with some standard well-known norm-based generalization bounds of Neyshabur et al. (2015), Bartlett et al. (2017), and Neyshabur et al. (2017) used as a baseline. We also add a comparison with more modern baselines such as the Hessian-Based Generalization from Ju et al. (2022) and the original compression bounds from Arora et al. (2018) on heavily pruned models. Our experiments compare the generalization error obtained by these bounds, the generalization bound of our algorithm (as provided in 5.2), and the true generalization error of the compressed, trained model. We also provide an experiment demonstrating how our generalization bound scales when increasing the hidden dimension of our models.

Our models are Multi-Layer Perceptron Models (MLPs) with ReLU activation layers with 5 layers. We train our algorithm with a learning rate of 0.02 with the Adam optimizer (Kingma & Ba, 2014) for 300 epochs. We conduct our experiments on two different image classification datasets: MNIST (LeCun & Cortes, 2010) and CIFAR10 (Krizhevsky et al.). We use an MLP with a hidden dimension of 784 to compare our bounds to other generalization bounds. For our experiments on scaling with model size, we test on hidden dimensions $500, 1000, 1500, 2000,$ and $2500$ where the depth is kept constant. For the comparisons with Arora et al. (2018) and Ju et al. (2022) only, we prune the model after training such that the only $\frac{1}{60}$ of the parameters[2] remain to demonstrate how our bounds capture sparsity better than existing bounds.

**Results** Our bounds are several orders of magnitude better than the baseline state-of-the-art generalization bounds, as can be inferred from Figure 3a and Figure 3b above. In both experiments, the closest bound to ours is that of Bartlett et al. (2017), which is still at least $10^3$ and $10^7$

|  | MNIST | CIFAR10 |
|---|---|---|
| Ours | 1.572 | 7.214 |
| Arora | 9.941 | 80.113 |
| Ju | 5.113 | 13.194 |

Table 1: On sparser models, our generalization bounds deliver tighter bounds than existing modern bounds.

---

[2]To note, we do not pick this number to cherry pick but instead pick ~~it from inspiration from Ma et al. (2019)~~.

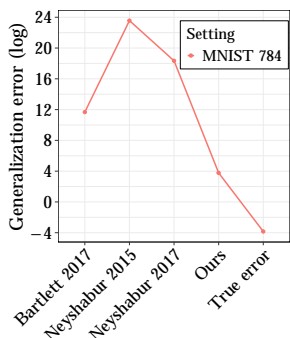 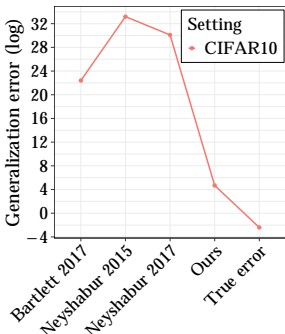 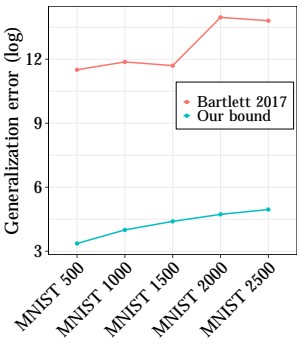

(a) Comparing bounds on MNIST.   (b) Comparing bounds on CIFAR10.   (c) Dependence on model size.

Figure 3: Comparison of the generalization bounds on logarithmic scale w.r.t. (a) MNIST, and (b) CIFAR10 datasets. In (c), we see how our bounds depend on the size of the model. It seems that our bounds grow relatively similarly to other norm-based bounds.

times greater than our bounds on the MNIST and the CIFAR10 datasets respectively. Moreover, our generalization bound is consistently better than Bartlett et al. (2017) as the hidden dimension of the models grows as in Figure 3c . This demonstrates that across several datasets, our bounds are tighter than traditional norm-based generalization bounds and scale better with hidden dimensions. Moreover, when using our generalization bounds on heavily pruned models, we see that our generalization bound also produces far tighter results than more modern bounds such as in Table 1. As is evident, the ability of our bounds to exploit the sparsity in the heavily pruned models yields tighter bounds than existing methodologies. To note, these numbers are on the standard scale, not the logarithmic scale. The results are not surprising mainly due to the use of our pruning algorithm, which ensures the error due to compression is low, and making use of Sparse Matrix Sketching, which significantly reduces the dimension of the pruned model, a key factor while computing generalization bounds.

## 8 CONCLUSION

This paper has made progress on the problem of deriving generalization bounds of overparametrized neural networks. We have obtained bounds for pruned models which are significantly better than well-known norm-based generalization bounds and empirically verified the effectiveness of this approach on actual data. We hope these results will fuel further research in deep learning to understand better how and why models generalize. It would be interesting to see if matrix sketching can be used to prove the generalization for different types of pruning, such as coresets. Moreover, it could also be fruitful to see whether matrix sketching can be used alongside PAC-Bayes bounds to yield generalization bounds as well. In this regard, we have extended the general framework of this paper to derive generalization bounds for lottery tickets in Section 6, a result which, to our knowledge, is the first of its kind. Another possibility would be to explore how different generalization bounds can be formed for different data distributions from different training algorithms.

**Limitations**   Our magnitude-based pruning algorithm does not prune the diagonal of the matrix, which is not standard. Moreover, after training, we assume that each atom belongs to an i.i.d Gaussian distribution, which may not always hold. Also, similar to the standard bounds, our generalization bounds are still vacuous, not fully explaining the generalization of the models.

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

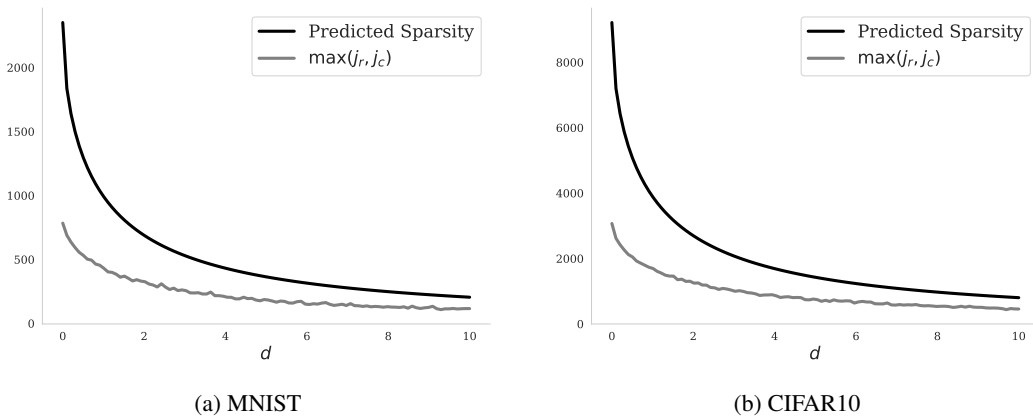

(a) MNIST              (b) CIFAR10

Figure 4: Here, we plot the predicted distribution-sparsity coefficients from Lemma 5.1 as well as the true in Lemma 5.1 holds empirically as well despite Assumption 3.1.

## A    COMPUTATION DETAILS

Here, we provide some information about the hardware and setup used for our work. We include a copy of the code in the Supplementary Material for reproducibility. Our experiments were run with an Intel(R) Xeon(R) Silver 4116 CPU @ 2.10GHz and an NVIDIA GeForce RTX 2080 Ti. Moreover, our experiments are run using Python version 3.10. Also, our experiments were done using a batch size of 256 for our experiments on MNIST and a batch size of 4 for CIFAR10. The Deep Learning software library used was PyTorch Lightning. We have additionally included our code in the supplementary material alongside with a README on how to reproduce our experiments.

## B    GRAPHICS OF DISTRIBUTED SPARSITY

**Empirical Verification of Sparsity Condition**    The proof of distributed sparsity is based on Assumption 3.1. However, we wish to empirically demonstrate that our claims on distributed sparsity still hold in practice and that Lemma 5.1 indeed holds empirically. We have trained MLPs on CIFAR10 and MNSIT and measured the value $\max(j_r, j_c)$ after pruning the trained MLPs. We have plotted how the true distributed sparsity value changes with the pruning parameter $d$ alongside our predicted upper bound of the distributed sparsity from Lemma 5.1. We plot this in Figure 4. We see that indeed, our error bound from Figure 4 indeed serves as an upper bound in practice despite the use of Assumption 3.1. For more experimental details, see Section 7.

## C    PROOF OF PRUNING AND DISCRETIZATION ERROR

We state that much of this error analysis is inspired by the proofs in Qian & Klabjan (2021). We, however, use the more reasonable Gaussian distribution assumption over the weights and have a slightly different Magnitude-based Pruning algorithm.

### C.1    PROOF OF LEMMA C.1

**Lemma C.1.** *The Expected Value of any entry $\Delta_{ij}^l$ of the matrix $\Delta^l = \hat{\mathbf{A}}^l - \mathbf{A}^l$ is given by $\mathbb{E}(\Delta_{ij}^l) = 0$ for any $i, j \in [d_1^l] \times [d_2^l], l \in [L]$. Thus, $\mathbb{E}(\Delta^l) = \mathbf{0}$ is a matrix full of $0$'s. Furthermore, we have that $\mathbb{E}((\Delta_{ij}^l)^2) = \frac{d^{\frac{3}{2}} \Psi}{(d+2)^{\frac{3}{2}}}$. Moreover, the fourth moment of any entry $(\Delta_{ij}^l)^4$ of $\Delta^l$ is given by $\mathbb{E}((\Delta_{ij}^l)^4) = \frac{3 d^{\frac{5}{2}} \Psi^2}{(d+2)^{\frac{5}{2}}}$.*

*Proof.* By the definition of our Algorithm 1, the random variable $\Delta_{ij}^l$ depends on the random variable $\mathbf{A}_{i,j}^l$.

$$\Delta_{ij}^l = \begin{cases} \mathbf{A}_{i,j}^l & \text{w.p. } \exp\left(\dfrac{-\left[\mathbf{A}_{i,j}^l\right]^2}{d\Psi}\right) \\ 0 & \text{w.p. } 1 - \exp\left(\dfrac{-\left[\mathbf{A}_{i,j}^l\right]^2}{d\Psi}\right) \end{cases}.$$

To calculate $\mathbb{E}\left(\Delta_{ij}^l\right)$, we will use the Law of Total Expectation. That is, $\mathbb{E}\left(\Delta_{ij}^l\right) = \mathbb{E}(\mathbb{E}(\Delta_{ij}^l|\mathbf{A}_{i,j}^l))$. We have that, $\mathbb{E}(\Delta_{ij}^l|\mathbf{A}_{i,j}^l) = \mathbf{A}_{i,j}^l \exp\left(\dfrac{-\left[\mathbf{A}_{i,j}^l\right]^2}{d\Psi}\right)$. Therefore, to calculate $\mathbb{E}(\mathbb{E}(\Delta_{ij}^l|\mathbf{A}_{i,j}^l))$, we use the definition of expectation for continuous variables as

$$\mathbb{E}(\Delta_{ij}^l) = \mathbb{E}(\mathbb{E}(\Delta_{ij}^l|\mathbf{A}_{i,j}^l))$$

$$= \int_{-\infty}^{\infty} \mathbf{A}_{i,j}^l \exp\left(\frac{-\left[\mathbf{A}_{i,j}^l\right]^2}{d\Psi}\right) \frac{1}{\sqrt{2\pi\Psi}} \exp\left(-\frac{1}{2}\left(\frac{\mathbf{A}_{i,j}^l}{\sqrt{\Psi}}\right)^2\right) d\mathbf{A}_{i,j}^l$$

$$= 0.$$

We now focus on the squared component of our lemma. Similarly, we have

$$\mathbb{E}((\Delta_{ij}^l)^2) = \mathbb{E}(\mathbb{E}((\Delta_{ij}^l)^2|\mathbf{A}_{i,j}^l))$$

$$= \int_{-\infty}^{\infty} (\mathbf{A}_{i,j}^l)^2 \exp\left(\frac{-\left[\mathbf{A}_{i,j}^l\right]^2}{d\Psi}\right) \frac{1}{\sqrt{2\pi\Psi}} \exp\left(-\frac{1}{2}\left(\frac{\mathbf{A}_{i,j}^l}{\sqrt{\Psi}}\right)^2\right) d\mathbf{A}_{i,j}^l$$

$$= \frac{d^{\frac{3}{2}}\Psi}{(d+2)^{\frac{3}{2}}}.$$

We similarly compute the fourth moment as

$$\mathbb{E}((\Delta_{ij}^l)^4) = \mathbb{E}(\mathbb{E}((\Delta_{ij}^l)^4|\mathbf{A}_{i,j}^l))$$

$$= \int_{-\infty}^{\infty} (\mathbf{A}_{i,j}^l)^4 \exp\left(\frac{-\left[\mathbf{A}_{i,j}^l\right]^2}{d\Psi}\right) \frac{1}{\sqrt{2\pi\Psi}} \exp\left(-\frac{1}{2}\left(\frac{\mathbf{A}_{i,j}^l}{\sqrt{\Psi}}\right)^2\right) d\mathbf{A}_{i,j}^l$$

$$= \frac{3d^{\frac{5}{2}}\Psi^2}{(d+2)^{\frac{5}{2}}}.$$

$\square$

## C.2 PROOF OF LEMMA C.2

**Lemma C.2.** *For any given layer $l \in [L]$, we have with probability at least $1 - \frac{1}{\epsilon_l}$*

$$\|(\hat{\mathbf{A}}^l - \mathbf{A}^l)\|_2 \le \epsilon_l \Gamma_l \quad \text{where} \quad \Gamma_l = C\left[\left(\sqrt{\frac{d^{\frac{3}{2}}\Psi}{(d+2)^{\frac{3}{2}}}}\right)\left(\sqrt{d_1^l} + \sqrt{d_2^l}\right) + \left(\frac{3d_1^l d_2^l d^{\frac{5}{2}}\Psi^2}{(d+2)^{\frac{5}{2}}}\right)^{\frac{1}{4}}\right].$$

*Here, $\hat{\mathbf{A}}^l$ is generated by Algorithm 1 and $C$ is a universal positive constant.*

We will prove that the error from the compression is bounded. To do so will first use the result that the expected norm of any zero-mean matrix can be bounded using the second and fourth moments. We restate this useful technical lemma from Theorem 2 of Latala (2005).

**Lemma C.3.** *Let $A$ be a random matrix whose entries $A_{i,j}$ are independent mean zero random variables with finite fourth moment. Then*

$$\mathbb{E}\|A\|_2 \le C\left[\max_i\left(\sum_j \mathbb{E}A_{i,j}^2\right)^{\frac{1}{2}} + \max_j\left(\sum_i \mathbb{E}A_{i,j}^2\right)^{\frac{1}{2}} + \left(\sum_j \mathbb{E}A_{i,j}^4\right)^{\frac{1}{4}}\right].$$

*Here, $C$, is a universal positive constant.*

We now use this lemma to bound the error due to compression using Algorithm 1.

*Proof.* Let $Z$ be our mask such that $\hat{\mathbf{A}}^l_{i,j} = Z \circ \mathbf{A}^l_{i,j}$ where $\circ$ is the elementwise-matrix product. We will analyze the difference matrix $\Delta = Z \circ \mathbf{A}^l_{i,j} - \mathbf{A}^l_{i,j}$. Note that

$$\mathbb{E}((\Delta^l_{ij})^2) = \mathbb{E}((\Delta^l_{ij})^2|Z_{i,j}=0) \cdot \mathbb{P}(Z_{i,j}=0) + \mathbb{E}((\Delta^l_{ij})^2|Z_{i,j}=1) \cdot \mathbb{P}(Z_{i,j}=1).$$

Trivially, if the mask for an atom is set to 1, the squared error for that atom is 0. Therefore, we have that

$$\mathbb{E}(\Delta^l_{ij}|Z_{i,j}=1)\mathbb{P}(Z_{i,j}=1) = 0.$$

Thus, we only need to analyze the second term. We have

$$\mathbb{E}((\Delta^l_{ij})^2|Z_{i,j}=0) \cdot \mathbb{P}(Z_{i,j}=0) = \mathbb{P}(Z_{i,j}=0)\int_{-\infty}^{\infty} (\mathbf{A}^l_{i,j})^2 \cdot \mathbb{P}(\mathbf{A}^l_{i,j}|Z_{i,j}=0)d\mathbf{A}^l_{i,j}$$

$$= \int_{-\infty}^{\infty} (\mathbf{A}^l_{i,j})^2 \cdot \mathbb{P}(Z_{i,j}=0|\mathbf{A}^l_{i,j}) \cdot \mathbb{P}(\mathbf{A}^l_{i,j})d\mathbf{A}^l_{i,j}$$

$$= \int_{-\infty}^{\infty} (\mathbf{A}^l_{i,j})^2 \cdot \exp\left(\frac{-[\mathbf{A}^l_{i,j}]^2}{d\Psi}\right) \cdot \frac{1}{\sqrt{2\pi\Psi}}\exp\left(-\frac{1}{2}\left(\frac{\mathbf{A}^l_{i,j}}{\sqrt{\Psi}}\right)^2\right)d\mathbf{A}^l_{i,j}$$

$$= \frac{d^{\frac{3}{2}}\Psi}{(d+2)^{\frac{3}{2}}}.$$

We then have that

$$\mathbb{E}((\Delta^l_{ij})^2) = \frac{d^{\frac{3}{2}}\Psi}{(d+2)^{\frac{3}{2}}}.$$

Similarly, we can do the same for the fourth moment.

$$\mathbb{E}((\Delta^l_{ij})^4|Z_{i,j}=0) \cdot \mathbb{P}(Z_{i,j}=0) = \mathbb{P}(Z_{i,j}=0)\int_{-\infty}^{\infty} (\mathbf{A}^l_{i,j})^4 \cdot \mathbb{P}(\mathbf{A}^l_{i,j}|Z_{i,j}=0)d\mathbf{A}^l_{i,j}$$

$$= \int_{-\infty}^{\infty} (\mathbf{A}^l_{i,j})^4 \cdot \mathbb{P}(Z_{i,j}=0|\mathbf{A}^l_{i,j}) \cdot \mathbb{P}(\mathbf{A}^l_{i,j})d\mathbf{A}^l_{i,j}$$

$$= \int_{-\infty}^{\infty} (\mathbf{A}^l_{i,j})^4 \cdot \mathbb{P}(Z_{i,j}=0|\mathbf{A}^l_{i,j}) \cdot \mathbb{P}(\mathbf{A}^l_{i,j})d\mathbf{A}^l_{i,j}$$

$$= \int_{-\infty}^{\infty} (\mathbf{A}^l_{i,j})^4 \cdot \exp\left(\frac{-[\mathbf{A}^l_{i,j}]^2}{d\Psi}\right) \cdot \frac{1}{\sqrt{2\pi\Psi}}\exp\left(-\frac{1}{2}\left(\frac{\mathbf{A}^l_{i,j}}{\sqrt{\Psi}}\right)^2\right)d\mathbf{A}^l_{i,j}$$

$$= \frac{3d^{\frac{5}{2}}\Psi^2}{(d+2)^{\frac{5}{2}}}.$$

Combining this with Lemma C.3, we have,

$$\mathbb{E}\|\Delta^l\|_2 \le C\left[\left(\sqrt{\frac{d^{\frac{3}{2}}\Psi}{(d+2)^{\frac{3}{2}}}}\right)\left(\sqrt{d^l_1} + \sqrt{d^l_2}\right) + \left(\frac{3d^l_1 d^l_2 d^{\frac{5}{2}}\Psi^2}{(d+2)^{\frac{5}{2}}}\right)^{\frac{1}{4}}\right].$$

We can then apply Markov's inequality where

$$\mathbb{P}(\|\Delta^l\|_2 \ge t) \le \frac{\mathbb{E}\|\Delta^l\|_2}{t}.$$

We set $\Gamma_l = C\left[\left(\sqrt{\frac{d^{\frac{3}{2}}\Psi}{(d+2)^{\frac{3}{2}}}}\right)\left(\sqrt{d^l_1} + \sqrt{d^l_2}\right) + \left(\frac{3d^l_1 d^l_2 d^{\frac{5}{2}}\Psi^2}{(d+2)^{\frac{5}{2}}}\right)^{\frac{1}{4}}\right]$ for notational ease. If we set $t = \epsilon_l\Gamma_l$, then we have with probability at least $1 - \frac{1}{\epsilon_l}$ that,

$$\|\Delta^l\|_2 \le \epsilon_l\Gamma_l.$$

We have proven our statement. □

### C.3 PROOF OF TRUE PERTURBATION BOUND

**Lemma C.4.** *For the weights of the model* $\mathbf{M}$ *and any perturbation* $\mathbf{U}^l$ *for* $l \in [h]$ *where the perturbed layer* $l$ *is* $\mathbf{U}^l + \mathbf{A}^l$, *given that* $\|\mathbf{U}^l\|_2 \leq \frac{1}{L}\|\mathbf{A}^l\|_2$, *we have that for all input* $x^0 \in \mathcal{S}$,

$$\|x^L - \bar{x}^L\|_2 \leq e\|x\|_2 \left(\prod_{l=1}^{L} \kappa_l L_l \|\mathbf{A}^l\|_2\right) \sum_{l=1}^{L} \frac{\|\mathbf{U}^l\|_2}{\|\mathbf{A}^l\|_2}.$$

*Here,* $\bar{x}^L$ *denotes the output of the Lth layer of the perturbed model.*

*Proof.* This proof mainly follows from Neyshabur et al. (2017). We restate it here with the differing notation for clarity and completeness. We will prove the induction hypothesis that $\|\bar{x}^l - x^l\|_2 \leq \left(1 + \frac{1}{L}\right)^l \|x^0\|_2 \left(\prod_{i=1}^{l} L_i\|\mathbf{A}^l\|_2\right) \sum_{i=1}^{l} \frac{\|\mathbf{U}^i\|_2}{\|\mathbf{A}^i\|_2}$. The base case of induction trivially holds, given we have that $\|\bar{x}^0 - x^0\|_2 = 0$ by definition. Now, we prove the induction step. Assume that the induction hypothesis holds for $l$. We will prove that it holds for $l + 1$. We have that

$$
\begin{aligned}
\|x^l - \bar{x}^l\|_2 &\leq \|\left(\mathbf{A}^l + \mathbf{U}^l\right)\phi_l(\bar{x}^{l-1}) - \mathbf{A}^l\phi_l(x^{l-1})\|_2 \\
&\leq \|\left(\mathbf{A}^l + \mathbf{U}^l\right)(\phi_l(\bar{x}^{l-1}) - \phi_l(x^{l-1})) + \mathbf{U}^l\phi_l(x^{l-1})\|_2 \\
&\leq \left(\|\mathbf{A}^l\|_2 + \|\mathbf{U}^l\|_2\right)\|\phi_l(\bar{x}^{l-1}) - \phi_l(x^{l-1})\|_2 + \|\mathbf{U}^l\|_2\|\phi_l(x^{l-1})\|_2 \qquad (2) \\
&\leq \left(\|\mathbf{A}^l\|_2 + \|\mathbf{U}^l\|_2\right)\|\phi_l(\bar{x}^{l-1}) - \phi_l(x^{l-1})\|_2 + \|\mathbf{U}^l\|_2\|\phi_l(x^{l-1})\|_2 \\
&\leq L_l\left(\|\mathbf{A}^l\|_2 + \|\mathbf{U}^l\|_2\right)\|\bar{x}^{l-1} - x^{l-1}\|_2 + L_l\|\mathbf{U}^l\|_2\|x^{l-1}\|_2 \qquad (3) \\
&\leq L_l\left(1 + \frac{1}{d}\right)\left(\|\mathbf{A}^l\|_2\right)\|\bar{x}^{l-1} - x^{l-1}\|_2 + L_l\|\mathbf{U}^l\|_2\|x^{l-1}\|_2 \\
&\leq L_l\left(1 + \frac{1}{d}\right)\left(\|\mathbf{A}^l\|_2\right)\left(1 + \frac{1}{L}\right)^{l-1}\|x^0\|_2 \left(\prod_{i=1}^{l-1} L_i\|\mathbf{A}^i\|_2\right)\sum_{i=1}^{l-1}\frac{\|\mathbf{U}^i\|_2}{\|\mathbf{A}^i\|_2} + L_l\|\mathbf{U}^l\|_2\|x^{l-1}\|_2 \\
&\qquad\qquad\qquad\qquad (4) \\
&\leq L_l\left(1 + \frac{1}{L}\right)^l \left(\prod_{i=1}^{l-1} L_i\|\mathbf{A}^i\|_2\right)\sum_{i=1}^{l-1}\frac{\|\mathbf{U}^i\|_2}{\|\mathbf{A}^i\|_2}\|x^0\|_2 + L_l\|x^0\|_2\|\mathbf{U}^l\|_2 \prod_{i=1}^{l-1} L_i\|\mathbf{A}^i\|_2 \\
&\leq L_l\left(1 + \frac{1}{L}\right)^l \left(\prod_{i=1}^{l-1} L_i\|\mathbf{A}^i\|_2\right)\sum_{i=1}^{l-1}\frac{\|\mathbf{U}^i\|_2}{\|\mathbf{A}^i\|_2}\|x^0\|_2 + \|x^0\|_2\frac{\|\mathbf{U}^l\|_2}{\|\mathbf{A}^l\|_2}\prod_{i=1}^{l} L_i\|\mathbf{A}^i\|_2 \\
&\leq \left(1 + \frac{1}{L}\right)^l \left(\prod_{i=1}^{l} L_i\|\mathbf{A}^i\|_2\right)\sum_{i=1}^{l}\frac{\|\mathbf{U}^i\|_2}{\|\mathbf{A}^i\|_2}\|x^0\|_2
\end{aligned}
$$

Here, Equation (2) results from applying Lemma D.1. Equation (3) comes from the fact that $\phi_i$ is $L_i$-Lipschitz smooth and that $\phi_i(0) = 0$. Moreover, Equation (4) comes from applying the induction hypothesis. Therefore, we have proven the induction hypothesis for all layers. We now only need the fact that $\left(1 + \frac{1}{L}\right)^L \leq e$, and we have our final statement. $\qquad\square$

### C.4 PROOF OF LEMMA C.5

**Lemma C.5.** *The difference between outputs of the pruned model and the original model on any input* $x$ *is bounded by, with probability at least* $1 - \sum_i^L \epsilon_i^{-1}$,

$$\|\hat{x}^L - x^L\| \leq e\|x\|_2 \left(\prod_{l=1}^{L} L_l\|\mathbf{A}^l\|_2\right)\sum_{l=1}^{L}\frac{\epsilon_l\Gamma_l}{\|\mathbf{A}^l\|_2}.$$

*Proof.* We will compare the output of the original model $x^l$ with the output of the compressed model. We need the fact that $\frac{1}{L}\|\mathbf{A}^l\|_2 \geq \epsilon_l\Gamma_l \geq \|\mathbf{A}^l - \hat{\mathbf{A}}^l\|_2$. From Vershynin (2019), we have that

$\mathbb{E}(\frac{1}{L}\|\mathbf{A}^l\|_2) \geq \frac{1}{4L}\left(\sqrt{d_1^l} + \sqrt{d_2^l}\right)$, and $\epsilon_l \Gamma_l$ is smaller than this in expectation for sufficiently small $\epsilon_l$. Therefore, we can use Lemma C.4 and Lemma C.2. Thus we have the following

$$\|x^l - \hat{x}^l\|_2 \leq e\|x\|_2 \left(\prod_{l=1}^{L} L_l\|\mathbf{A}^l\|_2\right) \sum_{l=1}^{L} \frac{\|\mathbf{A}^l - \hat{\mathbf{A}}^l\|_2}{\|\mathbf{A}^l\|_2}$$
$$\leq e\|x\|_2 \left(\prod_{l=1}^{d} L_l\|\mathbf{A}^l\|_2\right) \sum_{l=1}^{L} \frac{\epsilon_l \Gamma_l}{\|\mathbf{A}^l\|_2}$$

$\square$

## C.5 Proof of Lemma 4.2

**Lemma 4.2.** *The norm of the difference between the pruned layer and the discretized layer is upper-bounded as $\|\tilde{\mathbf{A}}^l - \hat{\mathbf{A}}^l\|_2 \leq \rho_l J_l$ where $J_l$ is the number of nonzero parameters in $\hat{\mathbf{A}}^l$ ($J_l$ is used for brevity here and will be analyzed later). Denote $x^L$ as the output of the original model and $\tilde{x}^L$ as the output of the pruned and discretized model. With probability at least $1 - \sum_{l=1}^{L} \epsilon_l^{-1}$, given that the parameter $\rho_l$ for each layer is chosen such that $\rho_l \leq \frac{\frac{1}{L}\|\mathbf{A}^l\|_2 - \epsilon_l \Gamma_l}{J_l}$, we have that the error induced by both discretization and the pruning is bounded by*

$$\|x^L - \tilde{x}^L\|_2 \leq e\|x\|_2 \left(\prod_{l=1}^{L} L_l\|\mathbf{A}^l\|_2\right) \sum_{l=1}^{L} \frac{\epsilon_l \Gamma_l + \rho_l J_l}{\|\mathbf{A}^l\|_2}.$$

*Proof.* We will compare the output of the original model $x^l$ with the output of the compressed and discretized model $\tilde{x}^l$. To use the perturbation bound from Lemma C.4, we need that $\|\mathbf{A}^l - \tilde{\mathbf{A}}^l\|_2 \leq \frac{1}{L}\|\mathbf{A}^l\|_2$. For each layer, we can choose a discretization parameter to satisfy this. We demonstrate this in the following:

$$\|\mathbf{A}^l - \tilde{\mathbf{A}}^l\|_2 \leq \|\mathbf{A}^l - \hat{\mathbf{A}}^l\|_2 + \|\hat{\mathbf{A}}^l - \tilde{\mathbf{A}}^l\|_2$$
$$\leq \epsilon_l \Gamma_l + \rho_l J_l$$

Therefore, as long as we choose

$$\rho_l \leq \frac{\frac{1}{L}\|\mathbf{A}^l\|_2 - \epsilon_l \Gamma_l}{J_l},$$

we have our desired property. Therefore, using Lemma C.4, we have that

$$\|x^l - \tilde{x}^l\|_2 \leq e\|x\|_2 \left(\prod_{l=1}^{L} L_l\|\mathbf{A}^l\|_2\right) \sum_{l=1}^{h} \frac{\|\mathbf{A}^l - \tilde{\mathbf{A}}^l\|_2}{\|\mathbf{A}^l\|_2}$$
$$\leq e\|x\|_2 \left(\prod_{l=1}^{d} L_l \kappa_l\|\mathbf{A}^l\|_2\right) \sum_{l=1}^{L} \frac{\|\mathbf{A}^l - \hat{\mathbf{A}}^l\|_2 + \|\hat{\mathbf{A}}^l - \tilde{\mathbf{A}}^l\|_2}{\|\mathbf{A}^l\|_2}$$
$$\leq e\|x\|_2 \left(\prod_{l=1}^{d} L_l\|\mathbf{A}^l\|_2\right) \sum_{l=1}^{L} \frac{\epsilon_l \Gamma_l + \rho_l J_l}{\|\mathbf{A}^l\|_2}$$

This happens only if the event from Lemma C.2 occurs for every layer. Using the union bound, we know that this happens with probability at least $1 - \sum_{l}^{L} \epsilon_l^{-1}$ $\square$

## D Error Bound under Subgaussian Conditions

We can form tighter bounds by considering what the expected maximum of $(\hat{\mathbf{A}}^l - \mathbf{A}^l)x$ would be with high probability. If $d_2^l < d_1^l$, we observe that the matrix $\hat{\mathbf{A}}^l - \mathbf{A}^l$ has at most $d_2^l$ nonzero singular values. We need a Subgaussian condition assumption on our input to each layer to do this well to improve our bounds.

**Condition D.1.** *The input to each layer $l \in [L]$, belongs to a distribution $\mathcal{D}$, such that for some unit magnitude vector $v$ and an arbitrary vector $x$ sampled from $\mathcal{D}$ satisfy*

$$\mathbb{P}(\langle x, v \rangle \geq t) \leq ae^{-bt^2 d_1^l}.$$

*Here $a$ and $b$ are universal constants greater than $0$.*

It should be noted that Condition D.1 is often seen throughout the theory of High Dimensional Statistics. The uniform distribution over the unit sphere satisfies Condition D.1. Given this Condition D.1, we can bound the approximation error from significantly increasing in any given layer.

We want to do a bound on how much error the compression introduces on the margin of the training dataset. While traditional bounds assume worst-case blow-up, we can use the fact that vectors are roughly orthogonal in high-dimensional spaces.

**Lemma D.1.** *Suppose we are given a matrix $\mathbf{B}$ of size $d_1^l \times d_2^l$ where $d_1^l \geq d_2^l$ and $\mathcal{S}$ is a collection of vectors from a distribution satisfying Condition D.1. For any vector $x \in \mathcal{S}$, there exists constants $a, b$ such that*

$$\|\mathbf{B}x\|_2 \leq \sqrt{d_2^l t_l}\|\mathbf{B}\|_2\|x\|_2,$$

*with probability at least $1 - |\mathcal{S}|ae^{-bt_l^2 d_1^l}$. We will call $\kappa_l = \sqrt{d_2^l t_l}$ if $d_2^l \leq d_1^l$ and $\kappa_l = 1$ otherwise.*

*Proof.* We first decompose $\mathbf{B} = U\Sigma V$ using Singular Value Decomposition. Therefore, for any $x$ we have that,

$$\begin{aligned}
\|\mathbf{B}x\|_2 &= \|U\Sigma Vx\|_2 \\
&= \|\Sigma Vx\|_2 \\
&= \|\Sigma y\|_2.
\end{aligned}$$

The second equality comes from the fact that $U$ is unitary and norm-preserving, and the third equality comes from setting $y = Vx$. Now, if $x$ is some standard random normal vector, then $y$ too is a standard random normal vector. We also observe that $\Sigma$ is a diagonal matrix where only the first $d_2^l$ values are nonzero. We use the well-known identity that if $v$ is a vector with a magnitude of $1$,

$$\mathbb{P}(\langle v, y \rangle \geq t) \leq ae^{-bt^2 d_1^l}.$$

Here, $a$ and $b$ are global constants. Therefore, applying this inequality for the respective nonzero singular values in $\Sigma$, we have

$$\mathbb{P}\left(\|\Sigma y\|_2 \geq \sqrt{d_2^l t}\|\mathbf{B}\|_2\right) \leq ae^{-bt^2 d_1^l},$$

since $\|B\|_2$ is the maximum singular value. Applying the union bound for each element of $\mathcal{S}$, we have that for every element in $\mathcal{S}$

$$\|\mathbf{B}x\|_2 \leq \sqrt{d_2^l t}\|\mathbf{B}\|_2\|x\|_2,$$

with probability at least $1 - |\mathcal{S}|ae^{-bt^2 d_1^l}$. $\square$

**Lemma D.2.** *For the weights of the model $\mathbf{M}$ and any perturbation $\mathbf{U}^l$ for $l \in [h]$ where the perturbed layer $l$ is $\mathbf{U}^l + \mathbf{A}^l$, given that $\|\mathbf{U}^l\|_2 \leq \frac{1}{L}\|\mathbf{A}^l\|_2$, we have that for all input $x^0 \in \mathcal{S}$,*

$$\|x^L - \bar{x}^L\|_2 \leq e\|x\|_2 \left(\prod_{l=1}^{L} \kappa_l L_l \|\mathbf{A}^l\|_2\right) \sum_{l=1}^{L} \frac{\|\mathbf{U}^l\|_2}{\|\mathbf{A}^l\|_2}.$$

*Here, $\bar{x}^L$ denotes the output of the Lth layer of the perturbed model. This happens if Condition D.1 occurs.*

*Proof.* This proof mainly follows from Neyshabur et al. (2017). We restate it here with the differing notation for clarity and completeness. We will prove the induction hypothesis that $\|\bar{x}^l - x^l\|_2 \leq \left(1 + \frac{1}{L}\right)^l \|x^0\|_2 \left(\prod_{i=1}^{l} \kappa_i L_i \|\mathbf{A}^l\|_2\right) \sum_{i=1}^{l} \frac{\|\mathbf{U}^i\|_2}{\|\mathbf{A}^i\|_2}$. The base case of induction trivially holds, given

we have that $\|\bar{x}^0 - x^0\|_2 = 0$ by definition. Now, we prove the induction step. Assume that the induction hypothesis holds for $l$. We will prove that it holds for $l + 1$. We have that

$$
\begin{aligned}
\|x^l - \bar{x}^l\|_2 &\leq \|\left(\mathbf{A}^l + \mathbf{U}^l\right)\phi_l(\bar{x}^{l-1}) - \mathbf{A}^l\phi_l(x^{l-1})\|_2 \\
&\leq \|\left(\mathbf{A}^l + \mathbf{U}^l\right)(\phi_l(\bar{x}^{l-1}) - \phi_l(x^{l-1})) + \mathbf{U}^l\phi_l(x^{l-1})\|_2 \\
&\leq \left(\|\mathbf{A}^l\|_2 + \|\mathbf{U}^l\|_2\right)\|\phi_l(\bar{x}^{l-1}) - \phi_l(x^{l-1})\|_2 + \|\mathbf{U}^l\|_2\|\phi_l(x^{l-1})\|_2 \quad (5) \\
&\leq \left(\|\mathbf{A}^l\|_2 + \|\mathbf{U}^l\|_2\right)\|\phi_l(\bar{x}^{l-1}) - \phi_l(x^{l-1})\|_2 + \|\mathbf{U}^l\|_2\|\phi_l(x^{l-1})\|_2 \\
&\leq L_l\left(\|\mathbf{A}^l\|_2 + \|\mathbf{U}^l\|_2\right)\|\bar{x}^{l-1} - x^{l-1}\|_2 + L_l\|\mathbf{U}^l\|_2\|x^{l-1}\|_2 \quad (6) \\
&\leq L_l\left(1 + \frac{1}{d}\right)\left(\|\mathbf{A}^l\|_2\right)\|\bar{x}^{l-1} - x^{l-1}\|_2 + L_l\|\mathbf{U}^l\|_2\|x^{l-1}\|_2 \\
&\leq L_l\left(1 + \frac{1}{d}\right)\left(\|\mathbf{A}^l\|_2\right)\left(1 + \frac{1}{L}\right)^{l-1}\|x^0\|_2\left(\prod_{i=1}^{l-1}L_i\kappa_i\|\mathbf{A}^i\|_2\right)\sum_{i=1}^{l-1}\frac{\|\mathbf{U}^i\|_2}{\|\mathbf{A}^i\|_2} + L_l\|\mathbf{U}^l\|_2\|x^{l-1}\|_2 \\
&\quad (7) \\
&\leq L_l\left(1 + \frac{1}{L}\right)^l\left(\prod_{i=1}^{l-1}L_i\kappa_i\|\mathbf{A}^i\|_2\right)\sum_{i=1}^{l-1}\frac{\|\mathbf{U}^i\|_2}{\|\mathbf{A}^i\|_2}\|x^0\|_2 + L_l\|x^0\|_2\|\mathbf{U}^l\|_2\prod_{i=1}^{l-1}L_i\|\mathbf{A}^i\|_2 \\
&\leq L_l\left(1 + \frac{1}{L}\right)^l\left(\prod_{i=1}^{l-1}L_i\kappa_i\|\mathbf{A}^i\|_2\right)\sum_{i=1}^{l-1}\frac{\|\mathbf{U}^i\|_2}{\|\mathbf{A}^i\|_2}\|x^0\|_2 + \|x^0\|_2\frac{\|\mathbf{U}^l\|_2}{\|\mathbf{A}^l\|_2}\prod_{i=1}^{l}L_i\|\mathbf{A}^i\|_2 \\
&\leq \left(1 + \frac{1}{L}\right)^l\left(\prod_{i=1}^{l}L_i\kappa_i\|\mathbf{A}^i\|_2\right)\sum_{i=1}^{l}\frac{\|\mathbf{U}^i\|_2}{\|\mathbf{A}^i\|_2}\|x^0\|_2
\end{aligned}
$$

Here, Equation (5) results from applying Lemma D.1. Equation (6) comes from the fact that $\phi_i$ is $L_i$-Lipschitz smooth and that $\phi_i(0) = 0$. Moreover, Equation (7) comes from applying the induction hypothesis. Therefore, we have proven the induction hypothesis for all layers. We now only need the fact that $\left(1 + \frac{1}{L}\right)^L \leq e$, and we have our final statement. If Condition D.1 is not satisfied, we need only set $\kappa_l = 1$ for all $l \in [L]$ and the analysis will remain valid. $\square$

### D.1 PROOF OF LEMMA D.3

**Lemma D.3.** *The difference between outputs of the pruned model and the original model on any input $x$ is bounded by, with probability at least $1 - \left[\sum_i^L \epsilon_i^{-1} + |\mathcal{S}|ae^{-bt_l^2 d_1^l}\right]$,*

$$
\|\hat{x}^L - x^L\| \leq e\|x\|_2\left(\prod_{l=1}^{L}L_l\kappa_l\|\mathbf{A}^l\|_2\right)\sum_{l=1}^{L}\frac{\epsilon_l\Gamma_l}{\|\mathbf{A}^l\|_2}.
$$

*Here, $a, b$ are positive constants from the distribution of input data.*

*Proof.* We will compare the output of the original model $x^l$ with the output of the compressed model. We need the fact that $\frac{1}{L}\|\mathbf{A}^l\|_2 \geq \epsilon_l\Gamma_l \geq \|\mathbf{A}^l - \hat{\mathbf{A}}^l\|_2$. From Vershynin (2019), we have that $\mathbb{E}(\frac{1}{L}\|\mathbf{A}^l\|_2) \geq \frac{1}{4L}\left(\sqrt{d_1^l} + \sqrt{d_2^l}\right)$, and $\epsilon_l\Gamma_l$ is smaller than this in expectation for sufficiently small $\epsilon_l$. Therefore, we can use Lemma D.2. Thus we have the following

$$
\begin{aligned}
\|x^l - \hat{x}^l\|_2 &\leq e\|x\|_2\left(\prod_{l=1}^{L}L_l\kappa_l\|\mathbf{A}^l\|_2\right)\sum_{l=1}^{L}\frac{\|\mathbf{A}^l - \hat{\mathbf{A}}^l\|_2}{\|\mathbf{A}^l\|_2} \\
&\leq e\|x\|_2\left(\prod_{l=1}^{d}L_l\kappa_l\|\mathbf{A}^l\|_2\right)\sum_{l=1}^{L}\frac{\epsilon_l\Gamma_l}{\|\mathbf{A}^l\|_2}
\end{aligned}
$$

$\square$

**Lemma D.4.** *The norm of the difference between the pruned layer and the discretized layer is upper-bounded as $\|\tilde{\mathbf{A}}^l - \hat{\mathbf{A}}^l\|_2 \leq \rho_l J_l$ where $J_l$ is the number of nonzero parameters in $\hat{\mathbf{A}}^l$ ($J_l$ is used for brevity here and will be analyzed later). With probability at least $1 - \left[ \sum_{l=1}^L \epsilon_l^{-1} + |\mathcal{S}|ae^{-bt_l^2 d_1^l} \right]$, given that the parameter $\rho_l$ for each layer is chosen such that $\rho_l \leq \frac{\frac{1}{L}\|\mathbf{A}^l\|_2 - \epsilon_l \Gamma_l}{J_l}$, we have that the error induced by both discretization and the pruning is bounded by*

$$\|x^L - \tilde{x}^L\|_2 \leq e\|x\|_2 \left( \prod_{l=1}^L L_l \kappa_l \|\mathbf{A}^l\|_2 \right) \sum_{l=1}^L \frac{\epsilon_l \Gamma_l + \rho_l J_l}{\|\mathbf{A}^l\|_2}.$$

*Proof.* We will compare the output of the original model $x^l$ with the output of the compressed and discretized model $\tilde{x}^l$. To use the perturbation bound from Lemma D.2, we need that $\|\mathbf{A}^l - \tilde{\mathbf{A}}^l\|_2 \leq \frac{1}{L}\|\mathbf{A}^l\|_2$. For each layer, we can choose a discretization parameter to satisfy this. We demonstrate this in the following:

$$\|\mathbf{A}^l - \tilde{\mathbf{A}}^l\|_2 \leq \|\mathbf{A}^l - \hat{\mathbf{A}}^l\|_2 + \|\hat{\mathbf{A}}^l - \tilde{\mathbf{A}}^l\|_2$$
$$\leq \epsilon_l \Gamma_l + \rho_l J_l$$

Therefore, as long as we choose

$$\rho_l \leq \frac{\frac{1}{L}\|\mathbf{A}^l\|_2 - \epsilon_l \Gamma_l}{J_l},$$

we have our desired property. Therefore, using Lemma C.4, we have that

$$\|x^l - \tilde{x}^l\|_2 \leq e\|x\|_2 \left( \prod_{l=1}^L L_l \kappa_l \|\mathbf{A}^l\|_2 \right) \sum_{l=1}^h \frac{\|\mathbf{A}^l - \tilde{\mathbf{A}}^l\|_2}{\|\mathbf{A}^l\|_2}$$
$$\leq e\|x\|_2 \left( \prod_{l=1}^d L_l \kappa_l \|\mathbf{A}^l\|_2 \right) \sum_{l=1}^L \frac{\|\mathbf{A}^l - \hat{\mathbf{A}}^l\|_2 + \|\hat{\mathbf{A}}^l - \tilde{\mathbf{A}}^l\|_2}{\|\mathbf{A}^l\|_2}$$
$$\leq e\|x\|_2 \left( \prod_{l=1}^d L_l \kappa_l \|\mathbf{A}^l\|_2 \right) \sum_{l=1}^L \frac{\epsilon_l \Gamma_l + \rho_l J_l}{\|\mathbf{A}^l\|_2}$$

This happens only if the event from Lemma C.2 and Lemma D.1 occur for every layer. Using the union bound, we know that this happens with probability at least $1 - \left[ \sum_l^L \epsilon_l^{-1} - |\mathcal{S}|ae^{-bt_l^2 d_1^l} \right]$. $\square$

## E    NAIVE GENERALIZATION PROOFS

Given the Gaussian assumption, it is natural to count the number of possible outcomes of the compression algorithm by counting the number of possible configurations of nonzero atoms in any matrix and then counting the possible values each atom could take after quantization. We provide the generalization bound from this intuition.

**Lemma E.1.** *Using the counting arguments above yields a generalization bound*

$$R_0(g_A) \leq \hat{R}_\gamma(f) + \mathcal{O} \left( \sqrt{\frac{\sum_l \log\left( \binom{d_1^l d_2^l}{\alpha} \right) + \alpha \log \frac{1}{\rho_l}}{n}} \right).$$

*This holds when $d$ is chosen such that $\gamma \geq e\|x\|_2 \left( \prod_{l=1}^L L_l \|\mathbf{A}^l\|_2 \right) \sum_{l=1}^L \frac{\epsilon_l \Gamma_l + \rho_l J_l}{\|\mathbf{A}^l\|_2}$.*

We now provide the requisite knowledge to prove this bound. We first analyze a naive methodology for counting the number of possible outcomes from the learning algorithm and compression scheme. We first provide a slightly altered generalization bound to fit our use case better.

**Theorem E.1.** *If there are $J$ different parameterizations, the generalization error of a compression $g_a$ is, with probability at least $1 - \delta$,*

$$L_0(g_A) \leq \hat{L}_\gamma(f) + \sqrt{\frac{\ln\left(\sqrt{\frac{J}{\delta}}\right)}{n}}.$$

*Proof.* Most of this proof follows from Theorem 2.1 from Arora et al. (2018). For each $A \in \mathcal{A}$, the training loss $\hat{R}_0(g_A)$ is the average of $n$ i.i.d. Bernoulli Random variables with expected value equal to $R_0(g_A)$. Therefore, by standard Chernoff bounds, we have that,

$$\mathbb{P}(R_0(g_A) - \hat{R}_0(g_A) \geq \tau) \leq \exp(-2\tau^2 n).$$

Given $f$ is $(\gamma, \mathcal{S})$-compressible by $g_A$, we know the empirical margin loss of $g_A$ for margin 0 is less than the empirical margin loss of $f$ with margin $\gamma$, i.e. $\hat{R}_0(g_A) \leq \hat{R}_\gamma(f)$. Given there are $J$ different parameterizations, by union bound, with probability at least $1 - J\exp(-2\tau n)$, we have $R_0(g_A) \leq \tau + \hat{R}_0(g_A)$. Setting $J\exp(-2\tau n) = \delta$, we have $\tau = \sqrt{\frac{\ln\left(\sqrt{\frac{J}{\delta}}\right)}{n}}$. Therefore, with probability $1 - \delta$, we have

$$R_0(g_A) \leq \hat{R}_\gamma(f) + \sqrt{\frac{\ln\left(\sqrt{\frac{J}{\delta}}\right)}{n}}.$$

$\square$

Now, we can state the number of parameterizations achievable by our compression algorithm. If there are $d_1^l d_2^l$ elements in the matrix and $\alpha$ stays nonzero after compression, then there are $\binom{d_1^l d_2^l}{\alpha}$ total parameterizations for each layer. Moreover, within each parameterization, there are $r^\alpha$ ways to choose the values that each nonzero element takes given each of the $\alpha$ atoms can take $r$ values where $r = \mathcal{O}\left(\frac{1}{\rho_l}\right)$. We, therefore, need a bound on the number of elements that stay nonzero after pruning. We achieve this with the following two lemmas. We will first prove that at least $\tau$ elements have probability $\kappa$ of getting compressed. Using such a counting argument yields the following generalization bound.

**Lemma E.2.** *At least $\tau$ elements of a given matrix $\mathbf{A}^l$ will have a probability of at least $\kappa$ of getting compressed. This event occurs with probability at least $1 - I_{1-p_1}(d_1 d_2 - \tau, 1 + \tau)$ where $p_1 = \mathrm{erf}\left(\sqrt{\frac{-d\ln(\kappa)}{2}}\right)$. Here, $\mathrm{erf}$ is the Gauss Error Function.*

*Proof.* For any given element to have a probability of at least $\kappa$ of getting compressed,

$$\exp\left(\frac{-\mathbf{A}_{i,j}^2}{d\Psi}\right) \geq \kappa.$$

This means that

$$|\mathbf{A}_{i,j}| \leq \sqrt{-d\Psi\ln(\kappa)}.$$

Given that $|\mathbf{A}_{i,j}|$ follows a Folded Normal Distribution, the probability of this happening is

$$p_1 = \mathbb{P}\left(|\mathbf{A}_{i,j}| \leq \sqrt{-d\Psi\ln(\kappa)}\right)$$

$$= \mathrm{erf}\left(\frac{\sqrt{-d\Psi\ln(\kappa)}}{\sqrt{2\Psi}}\right)$$

$$= \mathrm{erf}\left(\sqrt{\frac{-d\ln(\kappa)}{2}}\right) \tag{8}$$

For notational ease, we denote the set of atoms that satisfy this criterion $\mathcal{C} = \left\{ (i,j) | \exp\left(\frac{-\mathbf{A}_{i,j}^2}{d\Psi}\right) \geq \kappa \right\}$. Therefore, the number of elements $\tau$ that will have the probability of getting compressed larger than $\kappa$ obeys a binomial distribution. Therefore,

$$\mathbb{P}(|\mathcal{C}| \geq \tau) = 1 - I_{1-p_1}\left(d_1 d_2 - \tau, 1 + \tau\right).$$

Here, $I$ is the Regularized Incomplete Beta Function. □

Using this probabilistic upper bound from Lemma E.2, we can upper bound the number of nonzero elements in any matrix.

**Lemma E.3.** *Given the event from Lemma E.2 happens, the probability that at least $\alpha$ elements will end up being compressed is at least $1 - I_{1-\kappa}\left(\tau - \alpha, \alpha + 1\right)$.*

*Proof.* There are at least $\tau$ elements with probability greater than $\kappa$. In the worst case, the other $d_1^l d_2^l - \tau$ elements are not compressed. The probability distribution over the remaining elements is a binomial distribution with probability $\kappa$. Therefore, the probability that at least $\alpha$ of the $\tau$ elements are compressed is at least $1 - I_{1-\kappa}\left(\tau - \alpha, \alpha + 1\right)$. □

Now, we can finally prove our naive generalization bound.

*Proof.* From Theorem E.1, we have

$$L_0(g_A) \leq \hat{L}_\gamma(f) + \sqrt{\frac{\ln\left(\sqrt{\frac{J}{\delta}}\right)}{n}},$$

where $J$ is the number of parameter configurations. Each matrix has $\binom{d_1^l d_2^l}{\alpha}$ different ways to arrange the nonzero elements. Within any such configuration, there are $r^\alpha$ ways to select the values for any of the nonzero elements, where $r_l = \mathcal{O}\left(\frac{1}{\rho_l}\right)$ is the number of values any atom could take after discretization. This yields a generalization bound of

$$R_0(g_A) \leq \hat{R}_\gamma(f) + \mathcal{O}\left(\sqrt{\frac{\log(\binom{d_1^l d_2^l}{\alpha}) + \alpha \log r_l}{n}}\right).$$

Here, we only require that $\gamma \geq e\|x\|_2 \left(\prod_{l=1}^d L_l \kappa_l \|\mathbf{A}^l\|_2\right) \sum_{l=1}^L \frac{\epsilon_l \Gamma_l + \rho_l J_l}{\|\mathbf{A}^l\|_2}$ given Lemma D.4. □

## F  MATRIX SKETCHING PROOFS

### F.1  HOW TO CHOOSE $A, B$

To generate $A$ or $B$, we can first sample a random bipartite graph where the left partition is of size $m$, and the right partition is of size $p_1$ or the dimension of the matrix to be sketched. If we say that any node in the left partition is connected to at most $\delta$ nodes, we can call this bipartite graph a $\delta$-random bipartite ensemble. We have the resulting definition below.

**Definition F.1.** *$G$ is a bipartite graph $G = ([x], [y], \mathcal{E})$ where $x$ and $y$ are the size of the left and right partitions, respectively, and $\mathcal{E}$ is the set of edges. We call $G$ a $\delta$-random bipartite ensemble if every node in $[x]$ is connected to most $\delta$ nodes in $[y]$ and each possible connection is equally likely.*

Given this setup, we can choose the matrices $A$ and $B$ as adjacency matrices from a random $\delta$-random bipartite ensemble. Intuitively, such $A$ and $B$ are chosen such that any row in $A$ or $B$ has at most $\delta$ 1's. Any given element of $Y_{ij}$ is $\sum_{kl} A_{ik} \tilde{X}_{kl} B_{lj}$. However, only approximately $\delta^2$ of the elements in the sum are nonzero. Therefore, $Y_{ij}$ is expressed as the sum of $\delta^2$ terms from the sum $\sum_{kl} A_{ik} \tilde{X}_{kl} B_{lj}$. We can then express many elements from $Y$ by changing which elements are set or not set to zero in this sum. This gives a visual explanation of how this sketching works. Furthermore, the power of the expressiveness of the sketching depends mainly on the parameters $m$ and $\delta$. Here, we can bound the size required for $m$ and $\delta$ such that the solution to Equation (1) leads to one-to-one mapping with high probability.

### F.2 Remaining Proofs

Given that each of the atoms is identically distributed and independent, given $N$ atoms are not pruned, the problem of characterizing how these atoms are distributed among the rows or columns is similar to the famous balls-and-bins problem. We provide a helper lemma to prove that our pruning method generates a distributed sparse matrix. We use the famous lemma from Richa et al. (2000).

**Lemma F.1.** *Consider the problem of throwing $N$ balls independently a uniformly at random into n bins. Let $X_j$ be the random variable that counts the number of balls in the $j$-th bin. With probability at least $1 - n^{-\frac{1}{3}}$, we have that*

$$\max_j X_j \leq \frac{3N}{n}.$$

We now use this lemma to prove our distributed sparsity.

### F.3 Proof of Lemma 5.1

**Lemma 5.1.** *With probability at least $1 - \frac{1}{\lambda_l} - (d_1^l)^{-\frac{1}{3}} - (d_2^l)^{-\frac{1}{3}}$, we have that the outputs from Algorithm 1 are $j_r, j_c$-sparsely distributed where $\max(j_r, j_c) \leq 3\lambda_l \max(d_1^l, d_2^l)\chi$ and $\lambda_l \in \mathbb{R}$. Here, $\chi = \frac{\sqrt{d+2} - \sqrt{d}}{\sqrt{d+2}}$.*

*Proof.* We will first prove a bound on the number of noncompressed atoms, a random variable we will call $N$. The probability that any given element gets pruned is

$$\mathbb{P}(Z_{i,j} = 0) = \int_{-\infty}^{\infty} \mathbb{P}(Z_{i,j} = 0 | \mathbf{A}_{i,j}^l) \cdot \mathbb{P}(\mathbf{A}_{i,j}^l) d\mathbf{A}_{i,j}^l \tag{9}$$

$$= \int_{-\infty}^{\infty} \exp\left(\frac{-(\mathbf{A}_{i,j}^l)^2}{d\Psi}\right) \frac{1}{\sqrt{2\pi\Psi}} \exp\left(\frac{-1}{2} \frac{(\mathbf{A}_{i,j}^l)^2}{\Psi}\right) d\mathbf{A}_{i,j}^l \tag{10}$$

$$= \frac{\sqrt{d+2} - \sqrt{d}}{\sqrt{d+2}} \tag{11}$$

Therefore, the expected number of nonzero elements after pruning is $\mathbb{E}(N) = \frac{d_1^l d_2^l (\sqrt{d+2} - \sqrt{d})}{\sqrt{d+2}}$. Using Markov's inequality, we have that

$$\mathbb{P}(N \geq t) \leq \frac{\mathbb{E}(N)}{t}.$$

Here, we set $t = \lambda_i \frac{d_1^l d_2^l (\sqrt{d+2} - \sqrt{d})}{\sqrt{d+2}}$. Using this, we have with probability at least $1 - \frac{1}{\lambda_i}$,

$$N \leq \lambda_i \frac{d_1^l d_2^l (\sqrt{d+2} - \sqrt{d})}{\sqrt{d+2}}.$$

Here, we can use Lemma F.1. For the rows, with probability at least $1 - (d_1^l)^{\frac{-1}{3}}$, we have that the maximum number of nonpruned atoms in any row is at most

$$\frac{3N}{d_1^l} = 3\lambda_i \frac{d_2^l (\sqrt{d+2} - \sqrt{d})}{\sqrt{d+2}}.$$

Similarly, we have that the maximum number of nonpruned atoms in any column is at most

$$\frac{3N}{d_2^l} = 3\lambda_i \frac{d_1^l (\sqrt{d+2} - \sqrt{d})}{\sqrt{d+2}}.$$

Therefore, we have that this occurs with probability at least $1 - \frac{1}{\lambda_i} - (d_1^l)^{-\frac{1}{3}} - (d_2^l)^{-\frac{1}{3}}$. □

## F.4 PROOF OF THEOREM 5.2

**Theorem 5.2.** *For every matrix $\hat{\mathbf{A}}^l$, define $j_l$ to be the $\max(j_r, j_c)$ where $j_r$ and $j_c$ are the distribution-sparsity coefficients for $\hat{\mathbf{A}}^l$. Moreover, for every matrix $\hat{\mathbf{A}}^l$, define $p_l = \max(d_1^l, d_2^l)$. Then, we have that*

$$R_0(g_A) \leq \hat{R}_\gamma(f) + \mathcal{O}\left(\sqrt{\frac{\sum_l 3\lambda_l \chi d_2^l d_1^l \log^2(p_l) \log(\frac{1}{\rho_l})}{n}}\right).$$

*This holds when $d$ is chosen such that $\gamma \geq e\|x\|_2 \left(\prod_{l=1}^d L_l \|\mathbf{A}^l\|_2\right) \sum_{l=1}^L \frac{\epsilon_l \Gamma_l + \rho_l J_l}{\|\mathbf{A}^l\|_2}$ where $J_l \leq \mathcal{O}\left(\chi d_2^l d_1^l\right)$. This claim holds with probability at least $1 - \left[\sum_{l=1}^L \lambda_l^{-1} + \epsilon_l^{-1} + p_l^{-c}\right]$.*

*Proof.* From Lemma 5.1, we know that $\max(j_r, j_c) \leq 3\lambda_i \frac{\max(d_2^l, d_1^l)(\sqrt{d+2} - \sqrt{d})}{\sqrt{d+2}}$. Therefore, we can compress any matrix $\mathbf{A}^l$ into a sparse matrix $\hat{\mathbf{A}}^l$ and then further into a small matrix of size $(\sqrt{j_l p_l} \log(p_l))^2$ from Theorem 5.1. Therefore, we have that

$$(\sqrt{j_l p_l} \log(p_l))^2 \leq 3\lambda_i \frac{d_2^l d_1^l (\sqrt{d+2} - \sqrt{d})}{\sqrt{d+2}} \log^2(p_l).$$

By Theorem 3.1, we have that

$$L_0(g_A) \leq \hat{L}_\gamma(f) + \mathcal{O}\left(\sqrt{\frac{\sum_l 3\lambda_i \frac{d_2^l d_1^l(\sqrt{d+2}-\sqrt{d})}{\sqrt{d+2}} \log^2(p_l) \log(\frac{1}{\rho_l})}{n}}\right).$$

$\square$

## F.5 PROOF OF THEOREM 6.1

We present the error bound from Malach et al. (2020). This is formally presented in Theorem F.1.

**Theorem F.1.** *Fix some $\epsilon, \delta \in (0, 1)$. Let $\mathbf{M}$ be some target network of depth $L$ such that for every $i \in [L]$ we have $\|\mathbf{A}^i\|_2 \leq 1, \|\mathbf{A}^i\|_{\max} \leq \frac{1}{\sqrt{d_{1,i}}}$. Furthermore, let $n_\mathbf{M}$ be the maximum hidden dimension of $\mathbf{M}$. Let $G$ be a network where each of the hidden dimensions is upper bounded by $\text{poly}\left(d_{1,0}, n_\mathbf{M}, L, \frac{1}{\epsilon}, \log\frac{1}{\delta}\right) := D_G$ and depth $2L$, where we initialize $\mathbf{A}^i$ from the uniform distribution $U([-1, 1])$. Then, w.p at least $1 - \delta$ there exists a weight-subnetwork $\widetilde{G}$ of $G$ such that:*

$$\sup_{x \in \mathcal{S}} |\widetilde{G}(x) - \mathbf{M}(x)| \leq \epsilon.$$

*Furthermore, the number of active (nonzero) weights in $\widetilde{G}$ is $O\left(d_{0,1} D_G + D_G^2 L\right)$.*

**Theorem 6.1.** *Fix some $\epsilon, \delta \in (0, 1)$. Let $\mathbf{M}$ be some target network of depth $L$ such that for every $i \in [L]$ we have $\|\mathbf{A}^i\|_2 \leq 1, \|\mathbf{A}^i\|_{\max} \leq \frac{1}{\sqrt{d_{1,i}}}$. Furthermore, let $n_\mathbf{M}$ be the maximum hidden dimension of $\mathbf{M}$. Let $G$ be a network where each of the hidden dimensions is upper bounded by $\text{poly}\left(d_{1,0}, n_\mathbf{M}, L, \frac{1}{\epsilon}, \log\frac{1}{\delta}\right) := D_G$ and depth $2L$, where we initialize $\mathbf{A}^i$ from the uniform distribution $U([-1, 1])$. Moreover, with probability at least $1 - \delta - LD_G^{-c}$, there exists a weight-subnetwork $\tilde{G}$ of $G$ that obeys the generalization error of*

$$R_0(\tilde{G}) \leq \hat{R}_{\epsilon + \epsilon_\rho}(\tilde{G}) + \mathcal{O}\left(\sqrt{\frac{[n_\mathbf{M} d_{0,1} \log(D_G)^2 + L n_\mathbf{M}^2 \log(D_G)^2] \log\left(\frac{1}{\rho}\right)}{n}}\right).$$

*Here, $\epsilon_\rho$ is the small error introduced by discretization.*

*Proof.* Proving a generalization bound using our framework usually includes one, proving the error due to compression is bounded, and two, obtaining a bound on the number of parameters. Malach et al. (2020) fortunately proves both. We restate the bound from Arora et al. (2018):

$$R_0(\tilde{G}) \leq \hat{R}_\gamma(\tilde{G}) + \mathcal{O}\left(\sqrt{\frac{q \log r}{n}}\right).$$

From Theorem F.1, we have that

$$\sup_{x \in \mathcal{X}} |F(x) - \tilde{G}(x)| \leq \epsilon.$$

Directly setting $\gamma = \epsilon + \epsilon_\rho$ satisfies our error requirement, where $\epsilon_\rho$ is the small error introduced due to discretization. Now, we must focus on bounding the number of parameters in the model. Fortunately, Malach et al. (2020) provides a useful bound. They show that the first layer has approximately $\mathcal{O}(D_F d_0^1)$ nonzero parameters, and the rest of the layers of $\tilde{G}$ have approximately $\mathcal{O}(D_F^2)$ nonzero parameters. Moreover, from the proof of Theorem 2.1, they show that these nonzero parameters are evenly distributed across rows and columns. Therefore, we can use our matrix sketching framework to show that we can compress the set of outputs from Iterative Pruning to a smaller, dense set of matrices. Namely, the middle layers of $\tilde{G}$ such as $W_i^{\tilde{G}}$ can be represented as a smaller matrix of dimension $m = \mathcal{O}(D_F \log(D_G))$ from Theorem 5.1. For the first layer, we can also use matrix sketching to represent it as a matrix of size $\mathcal{O}(\sqrt{D_F d_{0,1}} \log(D_G))$. We now have an appropriate bound on the number of parameters in our sketched models. We apply trivial discretization by rounding the nearest value of $\rho$. Therefore, we have from Arora et al. (2018)

$$R_0(\tilde{G}) \leq \hat{R}_{\epsilon + \epsilon_\rho}(\tilde{G}) + \mathcal{O}\left(\sqrt{\frac{[D_F d_{0,1} \log(D_G)^2 + L D_F^2 \log(D_G)^2] \log\left(\frac{1}{\rho}\right)}{n}}\right).$$

We can apply the matrix sketching to each of the $L$ rows with probability at least $1 - D_G^{-c}$ according to Theorem 5.1. The error of the pruned model is also bounded by $\epsilon$ with at least probability $1 - \delta$. Union bounding these together show that this bound holds with probability at least $1 - \delta - L D_G^{-c}$. □

## G    ADDITIONAL EMPIRICAL RESULTS

We show the detailed empirical results on the MNIST and CIFAR10 datasets in Table 2 and Table 3 respectively, and are supplemental to the results obtained in Section 7. All bounds are shown on a logarithmic scale. We compare our bounds with some standard norm-based generalization bounds of Neyshabur et al. (2015),Bartlett et al. (2017), and Neyshabur et al. (2017). For comparing our bound on MNIST, we use an MLP with hidden dimensions 500, 784, 1000, 1500, 2000, and 2500 where the depth is kept constant. The model training details are detailed in Section 7. We see that across different hidden dimensions, our generalization bounds are consistently better than other generalization bounds. Over different hidden dimensions, the true generalization error seems to remain relatively stable. Relative to other bounds, our generalization bound seems more stable than other bounds, increasing at a lesser rate than other bounds as the hidden dimension increases. However, we unfortunately do not capture the seeming independence between the hidden dimension and true generalization error. For our bound, this is due to the fact that the margin of the trained model is not increasing enough with the increase in model size. Our bound predicts the generalization error of pruning in terms of the margin of the original model. If the margin of the original model does not increase while the model's size increases, our bound will increase. Therefore, this bound needs more information to capture generalization more accurately.

Additionally, we show the dependence of our bound on the number of training epochs in Figure 5, where we take the original MLP of depth 5 and compare how our generalization bound and the true generalization error change over epochs. It is to be noted that our bound is scaled to be in the same range as the true generalization error. There are differences between the curves, indicating our bound needs to include additional information needed to explain generalization fully. Our bound does decrease over time as the margin increases, mimicking the true generalization error. The main interesting difference is that the downward curve for our bound occurs in two places. The first drop

in our generalization bound happens only because of the drop of the generalization error, but the margin is still negative. Once the margin becomes positive and increases, our bound slowly begins to decrease. At this point, however, the true generalization error seems to have already reached its minimum.

In Table 3, all the insights noticed on the MNIST dataset seem to extend to CIFAR10. Our generalization bound is tighter than existing state–of–the–art norm-based generalization bounds. Indeed our error is orders of magnitude tighter than other generalization bounds. We note that while all generalization bounds here are far worse than the MNIST counterparts, our generalization bound most accurately reflects the true jump in generalization error between MNIST and CIFAR10. For both ours and the true generalilzatiion error, the bounds differ by one order of magnitude between MNIST and CIFAR10. However, the other bounds differ by at least 7 orders of magnitude. Our bound seems to capture more of the behavior of the true generalization error than these other bounds in this regard.

| METHOD | MNIST 500 | MNIST 784 | MNIST 1000 | MNIST 1500 | MNIST 2000 | MNIST 2500 |
|---|---|---|---|---|---|---|
| NEYSHABUR 2015 | 22.29 | 23.56 | 24.42 | 25.12 | 27.03 | 27.72 |
| NEYSHABUR 2017 | 17.91 | 18.34 | 18.70 | 18.81 | 21.50 | 21.57 |
| BARTLETT 2017 | 11.51 | 11.68 | 11.87 | 11.70 | 13.96 | 13.81 |
| OURS | 3.36 | 3.77 | 4.00 | 4.40 | 4.73 | 4.96 |
| TRUE ERROR | -3.76 | -3.84 | -3.80 | -3.85 | -3.86 | -3.87 |

Table 2: Generalization bounds on logarithmic scale w.r.t. MNIST using MLP of varying dimensions.

| METHOD | CIFAR10 |
|---|---|
| NEYSHABUR 2015 | 33.19 |
| NEYSHABUR 2017 | 30.10 |
| BARTLETT 2017 | 22.40 |
| OURS | 4.68 |
| TRUE ERROR | -2.41 |

Table 3: Comparison of different generalization bounds on the CIFAR10 dataset on a logarithmic scale

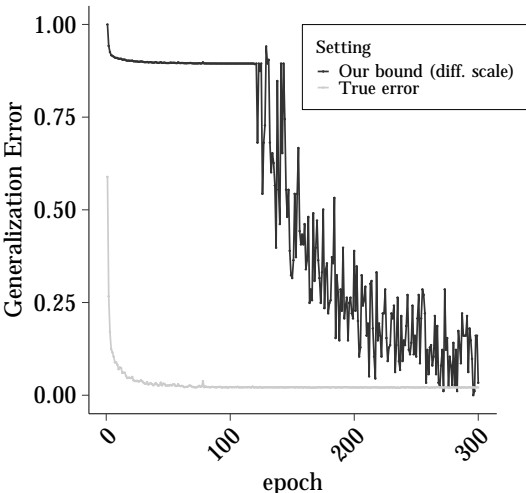

Figure 5: Comparing bounds on MNIST.

## H    EXTENSIONS TO CONVOLUTIONAL NETWORKS

Here, we add a short extension to Convolutional Networks. Here, we rely more strongly on the analysis from Qian & Klabjan (2021). There, they take a slightly different uniform assumption over the weights. We refer to Qian & Klabjan (2021) for details. However, they provide a result in the pruning error of CNNs which we will utilize. We note that this bound is much weaker since filters have low dimensionality so the gains from matrix sketching are not strong.

**Theorem H.1.** *For a K layer CNN F where each layer is a matrix $F_k \in \mathbb{R}^{d_k \times d_{k-1} \times p_k \times p_k}$, if we prune $d_k^\alpha$ filters where $0 \leq \alpha \leq 1$ randomly to generate $g_A$, we yield the generalization bound for the pruned model*

$$R_0(g_A) \leq \hat{R}_\gamma(F) + \mathcal{O}\left(\sqrt{\frac{\sum_{k=1}^K p_k^2 d_{k-1} d_k^\alpha \log^2(p_k^2 d_k) \log(\frac{1}{\rho_l})}{n}}\right).$$

*This requires that $\gamma \geq p^{-\beta_1} L_k^{K-1} p_0 \sqrt{K} \left[ p^{-\beta_1} \left( p^{-\beta_1} + D^{-\beta_2} \right)^{K-2} - p^{-(K-1)\beta_1} \right]$. Here, $\beta_1$ and $\beta_2$ are positive constants less than 1, $p = \max(\{p_1, \ldots, p_K\})$ and $D = \max(\{d_1, \ldots, d_K\})$.*

*Proof.* We will first analyze the error of pruning a CNN $F$ generating $\hat{F}$. From Qian & Klabjan (2021), we know that if we perform random filter-wise pruning for each convolutional layer such that only $d_k - d_k^\alpha$ filters remain, the maximum error over the inputs is

$$\sup_{x \in \mathcal{C}} \|F(x) - \hat{F}(x)\|_2 \leq p^{-\beta_1} L_k^{K-1} p_0 \sqrt{K} \left[ p^{-\beta_1} \left( p^{-\beta_1} + D^{-\beta_2} \right)^{K-2} - p^{-(K-1)\beta_1} \right].$$

Here, the constants can be $\beta_1 = 1$ and $\beta_2 = \frac{\alpha}{4}$. Again, this result comes from Qian & Klabjan (2021). We now use matrix sketching to find how many effective parameters are in CNNs. Any layer in a CNN has $d_k \times d_{k-1}$ filters of size $p_k \times p_k$. Pruning is done filter by filter, so we cannot apply matrix sketching to each filter. Fortunately, Sedghi et al. (2018) proved that a convolutional layer $F_k$ can be represented as a linear layer $W_k$ which belongs to $W_k \in \mathbb{R}^{p_k^2 d_k \times p_k^2 d_{k-1}}$. In this manner, $W_k \text{vec}(x) = F_K x$ where $x$ is some matrix and vec is a flattening operation. For details on this representation, please see Sedghi et al. (2018). However, for this proof, we only need to know that pruning a $d_k^\alpha$ filters results in this $W_k$ matrix having column-wise distributed sparsity where there are at most $(d_k - d_k^\alpha)p^2$ nonzero elements in any column. Therefore, applying matrix sketching to this $W_k$ matrix allows us to represent this sparse $p_k^2 d_k \times p_k^2 d_{k-1}$ matrix as a dense $m \times m$ matrix where $m = p_k^2 \sqrt{(d_k - d_k^\alpha)d_{k-1}} \log(p_k^2 d_k)$. This result comes from Theorem 5.1 from our work. Using Theorem 3.1 directly yields

$$R_0(g_A) \leq \hat{R}_\gamma(F) + \mathcal{O}\left(\sqrt{\frac{\sum_{k=1}^K p_k^4 d_{k-1}(d_k - d_k^\alpha) \log^2(p_k^2 d_k) \log(\frac{1}{\rho_l})}{n}}\right).$$

This requires that $\gamma \geq p^{-\beta_1} L_k^{K-1} p_0 \sqrt{K} \left[ p^{-\beta_1} \left( p^{-\beta_1} + D^{-\beta_2} \right)^{K-2} - p^{-(K-1)\beta_1} \right]$. ☐

## I    ADDITIONAL GAUSSIAN VERIFICATION

We have added empirical verification of our main assumption in Figure 6. This includes visual verification of the distribution of weights on the second and third layers. We see, indeed, that the weights are approximately Gaussian, even on the intermediate and final layers. This was a 4 layer neural network. We note that the final layer is a slightly skewed to the right for both, looking slightly less Gaussian.

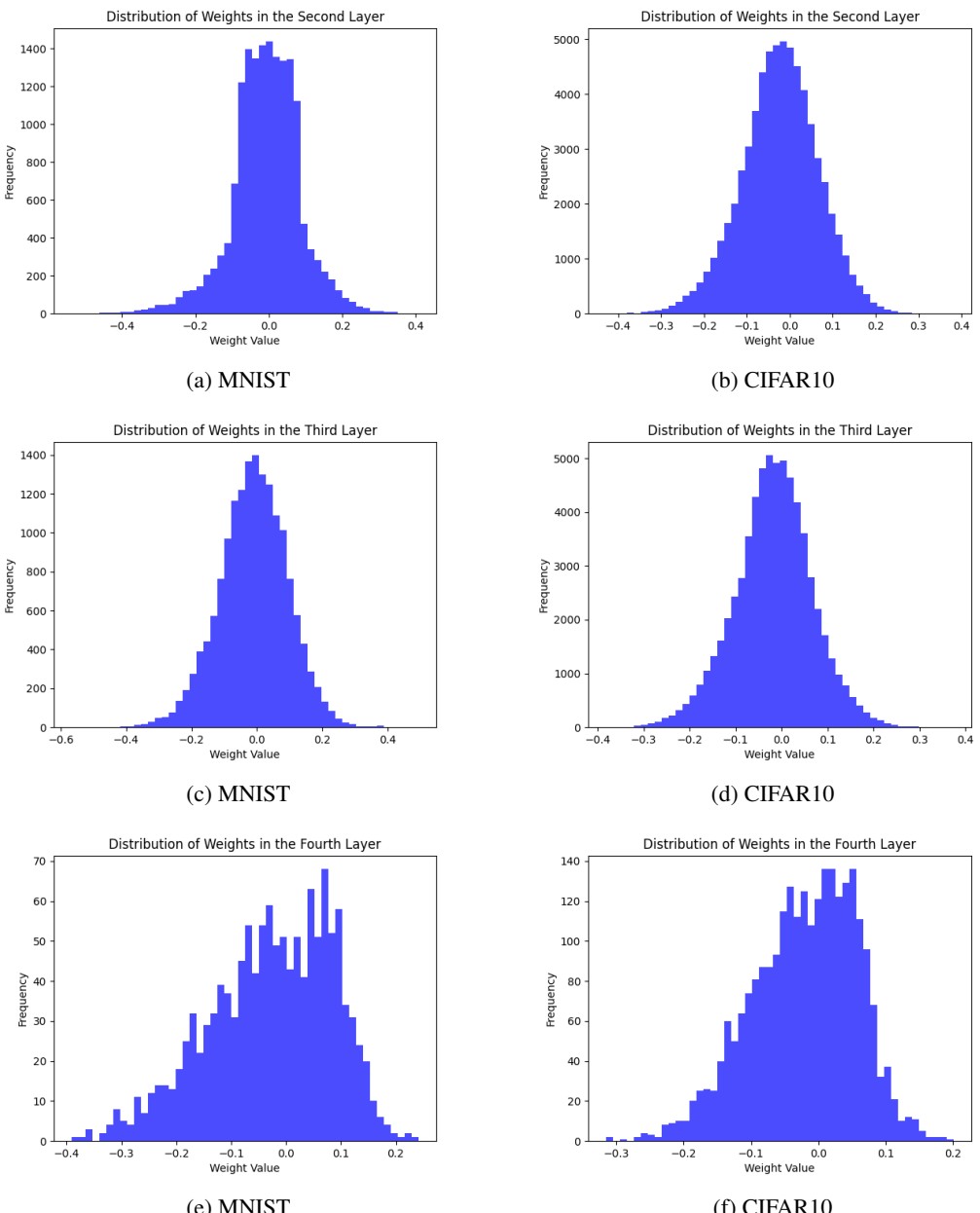

Figure 6: Here, we plot the empirical distribution over the weights to verify our Gaussian Assumption. This is on the second, third, and fourth layers of our models.

