# OpenReview forum: "Generalization Bounds for Magnitude-Based Pruning via Sparse Matrix Sketching"
_ICLR.cc/2024/Conference — Submitted to ICLR 2024_

### Official Review · Reviewer_v5dd · 2023-10-24

**Soundness:** 3 good
**Presentation:** 3 good
**Contribution:** 3 good
**Rating:** 6
**Confidence:** 3

**Summary:**

This paper studies the generalization bounds for neural networks with magnitude-based pruning. The approach is as follows. First, the paper proposes a magnitude-based pruning algorithm, which randomly sets the non-diagonal weights of neural networks to zero according to a Bernoulli random variable. This is followed by a discretization approach and a matrix sketching approach to improve the dependency of the bounds on the number of trainable parameters. The main result is a generalization bound based on a compression approach. Experimental analysis is also included to verify the assumptions used in the analysis, and to verify the theoretical results.

**Strengths:**

The paper proposed generalization bounds based on magnitude-based pruning, which does not require structured compression as considered in the existing studies.

The paper considers sparse matrix sketching to decrease the number of trainable parameters, which improves the generalization bounds.

**Weaknesses:**

Theorem 5.2 requires an assumption $\gamma$ to be no smaller than a number. It seems that this number would be very large since it has exponential dependency on $L$, and $\Gamma_l$ is also large according to the definition. If $\gamma$ is large, then $\hat{R}_\gamma$ would be very large.

To make the high-probability bound in Theorem 5.2 meaningful, one needs to choose very large $\epsilon_l,\lambda_l$ and $p_l$. In this case, the generalization bound would also be large. Furthermore, Lemma B.5 requires $\epsilon_l$ to be sufficiently small. This seems to lead to a contradiction.

In Theorem 5.2, the dependency of the generalization bound on $d$ seems to be $d^{3/4}$. If I understand correctly, if we do not use the matrix sketching, the dependency would be $d$. Therefore, the improvement is $d^{1/4}$, which seems not to be quite significant. As stated in the conclusion, the analysis still leads to a vacuous generalization bound.

In Lemma 5.1, the statement holds with probability $1-1/\lambda_1-d^{-1/3}$. This dependency on $d$ is not appealing since it would require a very large $d$ to get a bound holding with large probability. For example, if we want the bound to hold with probability $1-0.01$, then $d$ should be as large as $10^6$.

**Questions:**

Please see the comments above.

Minor comments:

Section 3.1: it seems that the definition of $\hat{R}_\gamma$ is not correct, i.e., $\geq \gamma$ should be $\leq\gamma$?

Theorem 3.1 only shows the existence of a $A\in\mathcal{A}$. How to find such a $A$?

Lemma 4.1: $\Gamma_l$ is used without giving its meaning

---

> ### Author Response · Authors · 2023-11-16
> **Author Rebuttal**
>
> We thank this reviewer for their thoughtful response! We have addressed their concerns here. If these satisfy your concerns, we would greatly appreciate an increase in score!
>
> # Weaknesses
> 1. *Theorem 5.2 requires an* …
>
> A. When $\gamma$ is large, $\hat{R}_{\gamma}$ is smaller as the acceptable margin is smaller. This is identical to the case of Arora et al. This is not an unusual statement unique to this paper.
>
> 2. *To make the high-probability bound in* …
>
> A. We don’t choose $p_l$, that is the maximum dimension of the matrix of the $l$th layer which can be large. The other two constants are pretty common constants in many generalization bounds. You see them in Arora et al. 2023 for example and more. This is not a weakness of this paper, rather a tradeoff of using concentration inequalities in general. Moreover, it is not the loosensess of these constants that makes the generalization bound big empirically. Moreover, it is not a contradiction. You just have to set $\epsilon$ accordingly. We found epsilon values which satisfy both empirically.
>
> 3. *In Theorem 5.2, the dependency of the* …
>
> A. This is not correct. Here, $d$ is a coefficient of the strength of the pruning. The dependence on $d$ is roughly $\frac{1}{d^{.25}}$ so as $d$ increases and prunes more, the generalization bound gets tighter and tighter. In every other generalization bound, there is no mention of this $d$ and they do not get tighter as you prune more. $d$ is not the dimension of the matrix. For heavily pruned matrices, our bounds are significantly tighter.
>
> 4. *In Lemma 5.1, the statement holds with* …
>
> A. This is incorrect. We understand the confusion and will fix this in the camera-ready version. $d$ is not $d_1^l, d_2^l$. Rather, $d$ is the pruning coefficient, while the latter is the dimension. In Lemma 5.1, these hold with probability $1 - 1/\lambda_1 - (d_1^l)^{-⅓} - (d_2^l)^{-⅓}$, which includes dependence on the matrix size. Therefore, this probability bound is strong in practice.
>
> # Minor comments:
>
> 1. *Section 3.1* …
>
> A. Yes, you are correct. This has been fixed in the updated version.
>
> 2. *Theorem 3.1* …
>
> A. Lemma 4.1 proves that using Algorithm 1 to prune a model finds such an $A$ that achieves a difference in output with small margin.
>
>
> 3. *Lemma 4.1* …
>
> A. We apologize for this. This was put into the appendix for space, but we will move it back for the Camera Ready Version.
>
> Sanjeev Arora, Rong Ge, Behnam Neyshabur, and Yi Zhang. Stronger generalization bounds for deep nets via a compression approach. CoRR, abs/1802.05296, 2018. URL http://arxiv.org/ abs/1802.05296.

---

### Official Review · Reviewer_TQ53 · 2023-10-24

**Soundness:** 2 fair
**Presentation:** 2 fair
**Contribution:** 3 good
**Rating:** 5
**Confidence:** 4

**Summary:**

In this paper, the authors present a generalization bound derived from magnitude-based pruning (MBP) sparsity.
The central idea is that if (a) the pruned model exhibits good generalization, and (b) the original model's performance closely matches the pruned model,
then the original model also generalizes well.
The validity of point (a) has been established by previous work (Arora et al.), so the primary focus of this paper is to substantiate point (b).
By assuming that magnitudes follow a Gaussian distribution, the authors demonstrate that pruned and discretized parameters closely approximate the original parameters. Furthermore, the authors introduce the concept of matrix sketching to reduce parameter size, leading to a more favorable bound. Empirical experiments support the authors' claims, showing that their bounds exceed test loss and outperform prior bounds.

Despite the paper's overall significance and the intuitiveness of the generalization bound based on MBP sparsity, there are still notable shortcomings. The most substantial concern lies in the limited ability to validate the proposed bound's practical effectiveness. The assumptions made, particularly regarding the Gaussian distribution of weights, lack robust justification in the experimental context. Additionally, the graphical verification of the bound may not be entirely convincing, as the predicted error bound substantially exceeds the actual error. This raises questions about the bound's validity and practical utility. Rather than focusing solely on the bound, it would be more insightful to verify the underlying assumptions.

Therefore, unfortunately, I cannot recommend an acceptance at the time being.

**Strengths:**

1. The authors introduce the intuitive concept of utilizing magnitude-based pruning sparsity to address generalization concerns.
2. Unlike prior approaches, this paper directly analyzes the generalization performance of the original model, rather than focusing solely on the pruned model.
3. The authors illustrate the adaptability of their technique to various contexts.

**Weaknesses:**

Overall, the proposed bound's practical effectiveness is challenging to confirm, representing a primary concern.

1. The assumptions, especially those concerning Gaussian weight distribution, lack adequate empirical validation.
2. The graphical verification of the bound might be misleading due to a significant predicted error bound compared to the actual error. A more rigorous approach is necessary to establish the validity of the assumptions.
3. Considering the assumed Gaussian distribution of weights, it is worth exploring whether better discretization methods exist.
4. The empirical verification of bounds in Fig~3 displays bound values that are excessively large, impairing their practical utility. It would be advisable to offer theoretical comparisons rather than relying solely on empirical evidence.

Minor Concerns:
This paper could significantly benefit from improved writing clarity. The current version may be confusing for readers. Some specific points to consider include:
In the Abstract:
1. Emphasize the necessity of Magnitude-Based Pruning and its advantages over structured pruning for clarity.
2. The use of "However" in the abstract seems to have unclear logic.
In the Introduction:
1. In the first paragraph, provide evidence supporting the claim that MBP significantly reduces memory requirements and inference time.
2. Consider rephrasing the use of "However" in the second paragraph for improved clarity.

**Questions:**

See limitations.

---

> ### Author Response · Authors · 2023-11-16
> **Official Response**
>
> We thank this reviewer for their thoughtful response! We have addressed their concerns here. If these satisfy your concerns, we would greatly appreciate an increase in score!
>
> # Weaknesses
> 1. *Overall, the proposed bound's* …
>
> A. There are two lemmas that use the Guassian assumption are Lemma 4.1 where pruning doesnt induce large error and that the sparsity is distributed. We verify empirically on MNIST and CIFAR10 that these are indeed the case. We verify the results of Lemma 4.1 in Figure 1 for example. Despite these assumptions being taken, we have verified that our lemmas do hold in practice. This is purely a weaker assumption than that of the Qian and Klabjan.
>
> 2. *The graphical verification of the bound* …
>
> A. We test both of the lemmas that come from our assumptions in Figures 1 and 2. We find that empirically, these bounds do indeed hold. Moreover, this Gaussian assumption is a purely weaker assumption than that done in Qian and Klabjan, which assume that the weights belong to a roughly uniform distribution. They take a stronger assumption to show a similar result that pruning does not induce large amounts of error.
>
> 3. *Considering the assumed Gaussian distribution* …
>
> A. We believe this reviewer has misinterpreted the point of discretization. We only discretize so that there are a limited number of parameterizations so that we can use the compression framework. Different discretization techniques are unlikely to improve the bounds significantly.
>
> 4. *The empirical verification of bounds*…
>
> A. As stated in this paper, all of the baselines do not capture the sparsity and do not incorporate sparsity into their generalization bounds. This means that our bounds are significantly tighter when the models are heavily pruned.
>
> ## Minor Concerns:
>
> A. We thank this reviewer for their comments on clarity. We have taken the time to improve these elements of our paper.
>
> Xin Qian and Diego Klabjan. A probabilistic approach to neural network pruning. CoRR, abs/2105.10065, 2021. URL https://arxiv.org/abs/2105.10065.

---

> > ### Comment · Reviewer_TQ53 · 2023-11-17
> > **Further clarification**
> >
> > I thank the authors for the timely responses.
> >
> > > There are two lemmas that use the Guassian assumption are Lemma 4.1 where pruning doesnt induce large error and that the sparsity is distributed. We verify empirically on MNIST and CIFAR10 that these are indeed the case. We verify the results of Lemma 4.1 in Figure 1 for example. Despite these assumptions being taken, we have verified that our lemmas do hold in practice.
> >
> > I believe that more experiments should be conducted to justify the Gaussian assumption.
> > For example, I suggest the authors *plotting the (empirical) weight distribution* and do some test to show that this distribution is, at least, close to Gaussian.
> >
> > As for the Fig 1, as I said before, the bound seems too large. So although this bound might be  larger than true generalization in practice, it cannot convince me that the Gaussian assumption is valid. For example, if I just propose a bound with constant 1000, this bound is also true but meaningless.
> >
> > Hope that this suggestion is helpful!

---

> ### Author Response · Authors · 2023-11-17
> **Response about Gaussian Assumption**
>
> Thank you for the responses! We now better understand your questions. We did not show this graphic since this is a well-known empirical phenomenon. For example, both Figure 7 by Han et al. 2015 and Figure 1 by Qian and Klabjan 2021 run experiments on the weight distributions of Feed-Forward Networks. After training, they *do* see that the distributions are indeed Gaussian approximately. They plot this graphical verification. We have reproduced their experiments and *added them to the newly uploaded version in Figure 1*. Please look at our figures. I believe these are what you are looking for. I appreciate all the help and quick response! Again, if this alleviates your concerns, we would greatly appreciate an increase in scores!
>
> Xin Qian and Diego Klabjan. A probabilistic approach to neural network pruning. CoRR,
> abs/2105.10065, 2021. URL https://arxiv.org/abs/2105.10065.
>
> Song Han, Jeff Pool, John Tran, and William Dally. Learning both weights and connections for
> efficient neural network. Advances in neural information processing systems, 28, 2015.

---

> > ### Comment · Reviewer_TQ53 · 2023-11-21
> > **Some more figures might be necessary**
> >
> > I thank the authors for updating Figure 1. However, Figure 1 only contains the first layer.
> > It cannot convince me that all the layers have the same performances.
> > Could the authors add some more figures (on other layers) about the Gaussian assumption in Appendix? I believe this could greatly enhance the current manuscript.

---

> ### Author Response · Authors · 2023-11-21
> **Response to Reviewer**
>
> We thank this reviewer for interacting with us. This is a reasonable request, and we have added 6 new plots in Appendix I in the newly uploaded version. As you can tell, the initial and intermediate layer weights look very Gaussian for all datasets. The last layer looks less Gaussian but still resembles a Gaussian, especially for CIFAR-10. Thus, we feel that a Gaussian distribution is still a reasonable assumption. We note that the mean over the distribution of weights of the last layer is roughly $-0.028$ for MNIST and $-0.021$ for CIFAR-10, so the assumption of zero-mean is still also reasonable. We appreciate the feedback and look forward to your response!

---

> > ### Comment · Reviewer_TQ53 · 2023-11-22
> >
> > I read the author's response and the other reviews. Based on the experiments the authors provide, I have become more natural to this paper.
> >
> > However, I still cannot find strong evidence that could let me recommend this paper with strong confidence, due to the current techniques and presentations.
> > This includes: (1) the techniques in this paper are not such novel, (2) the improvement for the bound seems not significant enough.
> > Besides, I suggest that the authors could do some statistical tests to see whether the weight indeed follows Gaussian, I cannot tell that it is Gaussian only based on my eyes.
> > So overall, I still lean negative but have become more natural. I changed my score from 3 to 5. Thanks again for the authors' response.

---

### Official Review · Reviewer_ftR9 · 2023-10-31

**Soundness:** 3 good
**Presentation:** 3 good
**Contribution:** 3 good
**Rating:** 6
**Confidence:** 3

**Summary:**

This paper aims to provide generalization bounds for pruned models using tools from sparse matrix sketching. The authors focus on developing a generalization bound for Magnitude based pruning methods but also show their proof methodology for other sparse subnetworks.

The proposed method first bounds the difference in layerwise activation norms between the dense and sparse model after pruning and discretization. The generalization bounds are then given by translating the sparse network into a small dense one using sparse matrix sketching and then applying bounds from Arora et al [1], which results in much tighter bounds.

**Strengths:**

1. The authors propose a novel idea of using sparse matrix sketching to develop generalization bounds for sparse networks.

2. Their establishes a connection between sparse networks obtained via magnitude pruning and generalization.

**Weaknesses:**

1. The proof follows a structure of bounding the layerwise error and then accounting for discretization of the parameters in order to conform with the setting of Theorem 3.1 based on the proof of Arora et al. [1]. However, the need for discretization has not been motivated clearly in the paper and can benefit from additional explanation regarding the same, which will make the proof easier to follow.

2. The magnitude pruning algorithm assumed by the authors uses a Bernoulli based construction of the sparse mask, however, in practice magnitude pruning is done based on sorting the parameters in each layer (or the entire network). Does the $(j_r, j_c)$ structure in each layer hold then? For pruning methods like Iterative Magnitude Pruning, larger layers are known to have dead neurons (corresponding to all zeros in a row). This problem is especially amplified in a CNN where IMP sets most channels to zero while having very few dense channels. The current proof seems to be unable to handle these situations and how will the assumption change for CNNs.

3. How will the generalization bounds change for different pruning methods, for eg: random pruning or other gradient based pruning criterions. Would random be strictly worse or similar? Insights on different pruning methods will help understand the relevance of the proposed bounds for different pruning criteria.

The key idea of the paper is novel and for the first time establishes generalization bounds for magnitude based pruning methods.

[1] Arora, Sanjeev, et al. "Stronger generalization bounds for deep nets via a compression approach." International Conference on Machine Learning.

**Questions:**

See above section for questions.

---

> ### Author Response · Authors · 2023-11-16
> **Official Response**
>
> We thank this reviewer for their thoughtful response! We have addressed their concerns here. If these satisfy your concerns, we would greatly appreciate an increase in score!
>
> # Weaknesses
> 1. *The proof follows a structure* …
>
> We apologize that this has not been made clear. We discretize so that the number of different parameterizations is finite. We have made this clearer in the uploaded new version.
>
> 2. *The magnitude pruning* …
>
> This is not true. There are different forms of Magnitude Based pruning. There are many forms of Magnitude Based pruning. For example, both random and threshold based Magnitude Based Pruning are discussed in Xian and Klabjan. This is not an abnormal form of pruning. However, our analysis can be slightly altered to prove for threshold based pruning as well. We only need that the pruning does not induce large error, which is known, and that there does not exist rows with many nonzeros, which we can also demonstrate. I am unsure what the second part of this weakness means. We only require that there are no rows with many nonzeros, i.e. distributed. Our analysis will hold if there are rows with no nonzeros.
>
> 3. *How will the* …
>
> As long as one can prove that the pruning does not induce large error and that the sparsity induced is distributed, our analysis will hold. In fact, our method is not largely dependent on the type of pruning used, only that it is evenly distributed and does not induce large errors. This makes our results much more generally applicable.
>
>
> Xin Qian and Diego Klabjan. A probabilistic approach to neural network pruning. CoRR, abs/2105.10065, 2021. URL https://arxiv.org/abs/2105.10065.

---

> > ### Comment · Reviewer_ftR9 · 2023-11-21
> > **Response to Rebuttal**
> >
> > I thank the authors for their clarifications on the pruning criterion, discretization and the sparsity pattern.
> >
> > From the discussion with Reviewer zSMw, If I understand correctly, the simple parameter counting argument achieves similar bounds to sketching and one could completely get rid of the sketching mechanism.
> >
> > Furthermore, the authors suggested that as long as the pruning ensures low induced error and no rows with many nonzeros, their analysis holds for any pruning criteria, not just magnitude pruning. The generalization argument for sparse networks thus follows discretization and parameter counting followed by using the results by Arora et al [1].
> >
> > It seems that the main contribution of the paper is then to combine the the compression result of Arora et al [1] with parameter counting for sparse networks, but this parameter counting too has been done partially by the Lottery Ticket kind of proofs by Malach et al [2], Pensia et al [3] and Burkholz [4].
> >
> > However, I still believe that providing a generalization bound for sparse networks is an interesting contribution, but the authors must appropriately modify their writing to highlight their exact contribution.
> >
> > [1] Arora, Sanjeev, et al. "Stronger generalization bounds for deep nets via a compression approach." International Conference on Machine Learning. 2018.
> >
> > [2] Malach, Eran, et al. "Proving the lottery ticket hypothesis: Pruning is all you need." International Conference on Machine Learning. PMLR, 2020.
> >
> > [3] Pensia, Ankit, et al. "Optimal lottery tickets via subset sum: Logarithmic over-parameterization is sufficient." Advances in neural information processing systems. 2020.
> >
> > [4] Burkholz, Rebekka. "Most activation functions can win the lottery without excessive depth." Advances in Neural Information Processing Systems. 2022.

---

> > > ### Author Response · Authors · 2023-11-21
> > > **Response to Reviewer**
> > >
> > > We thank this reviewer for the helpful comments. We have uploaded a new version of the paper with the contributions more explicitly specified in highlighted green.
> > >
> > > *We have specified our contribution relative to existing generalization bounds in the introduction*: "While analyses detailing pruning error bounds and parameter counting for pruned networks have been done before, they have not been utilized to deliver generalization bounds for pruned matrices. Moreover, we emphasize that while generalization bounds exist for sparse models, the arguments in these papers assume structure to the sparsity not existing in pruned models. We highlight that our analyses can extend to other forms of pruning that have low error and have sparsity evenly distributed throughout the matrix. For example, we extend our analysis to prove the generalization bounds of Lottery Tickets."
> > >
> > > *We have noted that both parameter counting techniques presented in the paper get good results*: "Moreover, using counting techniques or matrix sketching techniques yields efficient methods to count the number of efficient parameters in pruned weight matrices. Both simple counting techniques and sparse matrix sketching techniques achieve tight bounds and we present both styles of analysis in this paper."
> > >
> > > *We have also connected the relation to existing parameter counting techniques in the related works*: "We note that similar parameter counting techniques for Lottery Tickets are present in Burkholz (2022); Pensia et al. (2020)."
> > >
> > > We also note that the simple parameter counting argument in reference is *presented in the appendix*. It is already part of this paper. Both analysis techniques work and we showcase both of them. It is not the case that the simple counting argument was not present in this work.
> > >
> > > We greatly appreciate the help and we hope these edits clear up your concerns. If they do, we would greatly appreciate an increase in score. We appreciate all the help!

---

### Official Review · Reviewer_zSMw · 2023-11-02

**Soundness:** 1 poor
**Presentation:** 1 poor
**Contribution:** 2 fair
**Rating:** 1
**Confidence:** 2

**Summary:**

This paper proves new generalization bounds for deep neural networks via a compression based approach. They assume that the weights of the neural network are normally distributed and from this argue that a simple weight pruning approach well approximates the original network. They they argue that after pruning, the network can be represented with few parameters. This allows them to apply a generalization bound of Arora et al. 2018 which depends on the compression error and parameters of the compressed network.

The paper uses two key tools: (1) for their approximation step, they apply random matrix theory results to bound the error introduced by pruning (which due to their assumption of random weights is essentially an iid random perturbation of the weight matrix). (2) To bound the parameters of the pruned network, they argue that its weight matrices are sparse and thus can be compressed with a ‘sparse matrix sketching approach’ which basically allows recovering a sparse matrix X from a sketch AXB^T where A and B have many fewer rows than X.

Unfortunately, I found the results in the paper difficult to grasp — little intuition is given for the bounds and most of the results have undefined quantities which make them impossible to interpret. See my questions below. Further, it is not clear what role the two key technical tools actually have in establishing these bounds. Again, see my detailed questions below but: (1) it is not clear what the improvement is from applying random matrix theory bounds instead of simple Frobenius norm bounds to bound the spectral norm of th erandom perturbation (2) sparse matrix sketching does not represent a sparse matrix X with any fewer parameters than a naive sparse matrix format would. It is is clear that without additional assumptions, doing so would be impossible. Thus, it is unclear what role this tool is playing and why it is being used to bound the parameters of the compressed model.

**Strengths:**

See full review.

**Weaknesses:**

See full review.

**Questions:**

I have included comments/questions below. I starred points that I think are more important and would be helpful to have addressed during the author response period.

Questions/Comments:
- The abstract is confusing as it seems to focus on ML in general. Never mentions the context — e.g. neural networks.
- Overall, paper has lots of typos and grammatical errors. It would need significant proofing before final publication. The intro in particular is very vague and difficult to read. I could not understand from it what the paper was actually doing. E.g. what do you mean by sparse matrix sketching? From what I can tell, this is not a standard term and no citations are given in the intro when the term is introduced. Are you talking about random projection with sparse random matrices? Something else? How are you using this sparse matrix sketching? Is it an alternative to magnitude based pruning? A way of analyzing magnitude based pruning? Something else? Some of these questions become clearer later, but IMO the intro needs to be significantly reworked to be more concrete.
- In 3.1, what is the different between R(M) and R_0(M). Both notations are used and both seem to mean the case when gamma = 0? I.e. no margin?
-**  I didn’t understand the role of the ‘fixed string’ s in Def 3.1. It is not used in either the definition of G_{A,s} nor in the definition of compressibility. It seems that dropping it would not change the definition nor Theorem 3.1 at all. So what is its role? Relatedly, G_A,s is defined differently in Def 3.1 and Thm 3.1, which I presume is a typo.
- In Assumption 3.1, are the weights *independent* of each other? Or just marginally Gaussian but potentially correlated? (Later I see that independence is needed but this was not clear in the assumption)
- In Remark 4.1, what is meant by ‘diagonal elements’ given that the weight matrices are not square.
-** Given the explanation of the proof in 4.1, I’m not clear on why an assumption of Gaussianity was needed. And I’m not clear why randomized pruning was needed. As long as the original distribution was mean 0 and symmetric, if I just pruned all values below some threshold t, then Delta^l would have i.i.d. mean 0 entries with bounded moments right? And then random matrix theory results could be applied.
- In Lemma 4.1, what is Gamma_l? I couldn’t find where this was defined.
- ** I also couldn’t make intuitive sense of Lemma 4.1. I also don’t see how the bound could possibly not depend on some norm bound for the original input points. Say I just have 1 layer and the non-linearity is actually just the linear function, then if I multiply my input by some arbitrarily large constant C, then x^L and hat x^L will also be multiplied by C and thus the error would be scaled by C. So in this case, the error bound must depend on the scaling of the input points.
-** How does the error bound of Lemma 4.1 compared to the maximum size that ||x^L||_2 could be? Shouldn’t this maximum size be roughly proportional to Product L_l ||A^l||_2. Thus, wouldn’t this make the error bound very weak? As the error itself is large compared to ||x^L||_2?
- Fig 1 — is Lemma B.5 meant to refer to Lemma 4.1?
-** Where does the random matrix theory actually come in to Lemma 4.1? In particular, if I have a matrix with random mean 0 entries, then I can bound the spectral norm of that matrix trivially by the Frobenius norm, which by Markov’s inequality is at most something like n*E[Var(random variable)] which good probability. Random matrix theory results improve this bound to something scaling more like \sqrt{n}. If you instead used the trivial Frobenius norm bound, how would Lemma 4.1 change?
- In Section 5 it was unclear of the Sparse Matrix Sketching idea was novel to this paper, or something from the literature? No citations are given.
- **Say I have a p x p matrix with j*p nonzeros. Then to represent this matrix in sparse matrix format, I need j*p values for by non-zero entries, along with j*p indices indicating their positions. These indices take ~ log p bits each. Thus, I need roughly j*p*log p parameters to represent the whole matrix. I don’t see how sparse matrix sketching is improving on this simple argument. In fact, since it compressed to a \sqrt{jp}logp x \sqrt{jp} log p sized dense matrix, it seems to lose a log factor since (\sqrt{jp} log p)^2 = jp log^2 p parameters.
- ** Related to the above question, an alternative bound based on a simple counting argument is given in Lemma D.1. The claim is that this bound is ‘combinatorial’. But it is not, given that {d_1 d_2 choose alpha} is in a log and thus bounded by roughly alpha*log(d_1 d_2).  I could not make sense of the bound of Theorem 5.2 enough to compare them but I couldn’t understand why Lemma D.1 was obviously a worse bound. We should be able to bound the number of non-zero entries alpha directly using Lemma 5.1 by something like max(d_1,d_2)*max(j_r,j_c). Doing so looks like it would in fact give a stronger bound that Theorem 5.2
- In Lemma 5.1, what is lambda_l? Without this being defined, it is impossible to interpret the theorem. E.g. we could have lambda_l = max(d_1,d_2). Also what roughly is Chi?
- In Theorem 5.2, it says that d must be chosen to make some expression hold. However, that expression does not depend on d. So I don’t understand what this is saying.

---

> ### Author Response · Authors · 2023-11-16
> **Official Response Part 1**
>
> We appreciate this reviewer's thoughtful review. We feel there are some parts miscommunicated. We hope these answers alleviate your concerns and raise your rating of our paper.
>
> # Weaknesses
>
> 1. *The abstract is confusing as it seems*…
>
> A. Magnitude-based Pruning is a very specific term as it applies to neural networks. We hope this clears up the confusion.
>
>
> 2. *Overall, paper has lots of typos*…
>
> A. Matrix Sketching is a well-known technique in matrix theory. I have added the relevant citations in the main text. We use Matrix Sketching to represent the space of sparse matrices in the set of smaller dense matrices to create tight generalization bounds from “Specifically, we show that our set of pruned matrices can be efficiently represented in the space of dense matrices of much smaller dimensions via the Sparse Matrix Sketching.”  We have fixed this in the uploaded new version.
>
> 3. *In 3.1, what is the difference between R(M) and* …
>
> A. We state in the paper that “The population risk $R(M)$ is obtained as a special case of $R_{\gamma}(M)$ by setting $\gamma = 0.” The fixed string is mentioned as we are repeating the lemma exactly from the work of Arora, which has the flexibility to use such fixed strings, but we do not use them. We have fixed this in the newly uploaded version. Moreover, yes, this is a small typo. We have dropped the string notation, as mentioned in the newly uploaded version.
>
> 4. *In Assumption 3.1, are the weights independent of* …
>
> A. We agree with this comment. We have specified the need for independence in the newly uploaded version.
>
> 5. *In Remark 4.1, what is meant by ‘diagonal elements’ given that* …
>
> A. The diagonal elements of a nonsquare matrix $A$ refer to any element $A_{i, i}$ for any $i$. We have made this more evident in the uploaded text. This intuition is where we believe this reviewer has misinterpreted this result. We need two facts: the error induced by pruning is small, and the sparsity is evenly distributed throughout the weights. If we assume gaussianity, which has been noticed in the past according to Han, both hold. Moreover, we study why randomized pruning leads to generalization, an empirical phenomenon. Magnitude-based pruning can be removing elements below a threshold or removing elements randomly with smaller values that are more probable (see Qian and Klabjan 2021). We can extend our results to deterministic pruning, but we still need the sparsity to be roughly distributed for matrix sketching to work. Thus, it is likely that a Gaussian assumption is still necessary even for deterministic pruning. We also apologize for not including the definition of $\gamma_L$ in the main text; we moved its definition to the appendix for space reasons. We will move it back to the main text. We want to represent the possible space of parameterizations of sparse matrices as smaller dense matrices to generate tighter generalization bounds.

---

> > ### Comment · Reviewer_zSMw · 2023-11-16
> >
> > Thanks for the explanations.
> >
> > I still am confused about the Gaussianity assumption. What would break in the analysis if I used any other bounded distribution which was independent and identical across entries instead of Gaussian? The error would still be small and the sparsity would be evenly distributed by the iid assumption no?
> >
> > As a follow up question -- why would deterministic pruning not work if I have iid random weights? Again, even though the pruning threshold is deterministic, the pruned weights would be well-spread due to the random weight assumption.

---

> > > ### Author Response · Authors · 2023-11-16
> > > **Official Response**
> > >
> > > We thank the reviewer for the good questions. We hope these clear up your questions and you look upon our paper more favorably.
> > >
> > > 1. You have made a great observation. In fact, *we truly only need the weights' independence and identicality and a zero mean distribution. Any distribution satisfying this satisfies our analysis. For example, the uniform distribution from Qian and Klabjan would have also been simple to analyze and would work with our current analysis. We take the Gaussian assumption *only* for the reason that, empirically, weights tend to look Gaussian, not uniform. We can very simply extend our analysis to work on different distributions. If this reviewer believes it is better, we can remove the Gaussian assumption with an IID and zero mean and bounded variance distribution for the Camera Ready Version. It is a very simple change and does not affect our analysis at all.
> > >
> > > 2.  Our analysis *can* support deterministic pruning. You would want to assume the weights are i.i.d. random and zero mean. If we adopt the pruning analysis for deterministic pruning from Qian and Klabjan for deterministic pruning, using our generalization bound with deterministic pruning is easy. The only difference would be the proof that pruning does not induce large errors, but that is also very simple to prove. Extending to different forms of pruning is simple, hence why we extend our analysis for Lottery Tickets. If this reviewer wants, we can extend our analysis to deterministic pruning in the appendix for the camera ready. It is very simple to do.

---

> ### Author Response · Authors · 2023-11-16
> **Official Response Part 2**
>
> # Weaknesses
>
> 6. *I also couldn’t make intuitive sense of Lemma 4.1*…
>
> A. You are correct; there is a small typo in the proof. The norm of the input should scale with this error. We have corrected this in the newly uploaded version. This is a minor change. The latter part about the significant error compared to $\|x^L\|_2$ is invalid. As the error induced by pruning decreases, epsilon decreases, and the error decreases. The main point of this proof is that the error scales linearly with epsilon. This margin is not tight in practice, as seen in Figure 1a. This looseness is normal in margin bounds and suffices for our generalization bounds. This looseness and style of analysis are similar to the results of Arora et al. 2018. While one can trivially reduce this dependence using noise stability, such as Arora et al. 2018, this only slightly improves this bound.
>
>
> 7, *Fig 1 — is Lemma B.5 meant to refer to Lemma 4.1?* -...
>
> A. Yes, there are many ways to prove that pruning does not induce too much error. We agree with this reviewer that there are many ways to do this. Again, analyzing the spectral norm of the difference between the original and pruned matrix is a straightforward way to analyze this and follows some of the analysis of the Probalistic pruning change. We do not believe alternative methods to proving similar bounds make this standard way any less strong. Random Matrix Theory is useful for demonstrating that the spectral norm of the difference between a pruned matrix and the original matrix is relatively small. There are many ways to show this. However, trivially applying Markov’s Inequality results in a very poor bound. Using it results in an upper bound of $\|\hat{\mathbf{A}} - \mathbf{A}\|_2 \leq c\|\mathbf{A}\|_2$ where $c$ is some large constant. This bound is weaker than ours.
>
> 8. *In Section 5 it was unclear of the Sparse Matrix Sketching* …
>
> A. In Theorem 5.1, we cite Dasarthy et al. Matrix Sketching is an idea from existing papers, but its use for generalization bounds is indeed novel.
>
>
> 9. *Say I have a p x p matrix with jp nonzeros* …
>
> A. This is an equivalent argument to the combinatorial argument in the appendix. We agree with this reviewer. We thank you for this catch. Indeed, the argument in the appendix is just as tight as matrix sketching. However, the argument in the appendix is still strong and improves on existing bounds. We hope this is enough in your opinion for acceptance.
>
> 10. *Related to the above question*, …
>
> A. We agree with this reviewer. Please see above.
>
> 11. *In Lemma 5.1, what is lambda_l?* …
>
> A. As stated, Lemma 5.1 holds for any $\lambda$ in $\mathbb{R}$. Moreover, $\chi$ is exactly as stated in Lemma 5.1.
>
>
> 12. *In Theorem 5.2, it says that d must be chosen* …
>
> A. $\epsilon_l$ depends on d from the pruning error. This comes from Lemma 4.2.
>
>
> # References
>
> Xin Qian and Diego Klabjan. A probabilistic approach to neural network pruning. CoRR, abs/2105.10065, 2021. URL https://arxiv.org/abs/2105.10065.
>
> Sanjeev Arora, Rong Ge, Behnam Neyshabur, and Yi Zhang. Stronger generalization bounds for deep nets via a compression approach. CoRR, abs/1802.05296, 2018. URL http://arxiv.org/ abs/1802.05296.
>
> Gautam Dasarathy, Parikshit Shah, Badri Narayan Bhaskar, and Robert D. Nowak. Sketching sparse
> matrices. CoRR, abs/1303.6544, 2013. URL http://arxiv.org/abs/1303.6544.

---

> > ### Comment · Reviewer_zSMw · 2023-11-16
> >
> > Just to follow up: how exactly would Lemma 4.1 change if we just used a naive bound on ||A-tilde A||? From what I can tell, the theorem already has a large constant in its bound anyways.
> >
> > Am I correct that applying sparse matrix sketching then does not give better generalization bounds then naive counting arguments? If this is the case I am worried about how the paper is framed. It focuses on using random matrix theory and sparse matrix sketching as tools for generalization bounds. But it sounds as if there is no need for at least one of these tools.
> >
> > I also think the description in the paper is incorrect then. In particular, don't the following paragraphs of the paper contradict the above response?
> >
> > "Counting the number of parameters in small dense matrices
> > is more efficient in terms of parameters than counting the number of large sparse matrices, thus
> > providing a way of evasion of the combinatorial explosion. We formalize this in the following section."
> >
> > "Regrettably, such a bound is quite poor in its dependence on the size of the matrices, mainly due
> > to the logarithm term of the factorial, which is a significantly large value. This is, in fact, worse
> > than many of the previous bounds in the literature. This is due to the combinatorial explosion of the
> > number of sparse matrices. However, if there exists a way to instead represent the space of sparse
> > matrices within the space of dense matrices of much smaller dimensions, then we could avoid such a
> > combinatorial explosion of the number of parameters. This is the exact purpose of matrix sketching"

---

> > > ### Author Response · Authors · 2023-11-16
> > > **Official Response (Part 2)**
> > >
> > > We continue to respond to this reviewer's questions.
> > >
> > > 3. Lemma 4.1 would change to have a dependence on $d_1^ld_2^l \frac{d^\frac{3}{2}}{(d+2)^{\frac{3}{2}}}$ using your analysis. This bound is much looser than ours since ours is dependent on dimension roughly $\sqrt{d_1^l} + \sqrt{d_2^l}$. This is much better. We hope this tightness justifies our style of analysis.
> > >
> > > 4. We agree that the combinatorial argument in the appendix is similar in result to the matrix sketching one. We thank the reviewer for pointing this out. However, we argue that the point of this paper was to *deliver generalization bounds that consider the sparsity induced by pruning*. The matrix sketching and combinatorial ideas are the first to do these in the literature. We believe that this does not weaken the core result of our paper. This paper has demonstrated two ways to prove generalization bounds for pruned models. Changing the framing of the introduction is simple and can be done for the Camera-Ready version or even this uploaded version. We do not believe that the existence of an alternative analysis style achieving similar results that we thoroughly analyze in the appendix weakens this paper. However, we agree the framing has to be changed in the introduction and some parts of the main body. *We have already fixed the framing accordingly in the uploaded version.* We hope that these explanations and arguments help clear up any concerns the reviewer has.

---

### Author Response · Authors · 2023-11-22
**Message to all reviewers**

Dear Reviewers,

Thank you again for your detailed feedback.

We believe we have addressed the main concerns and invite you to engage with us with any remaining questions or unclear parts about our paper. If we have addressed your concerns, we would greatly appreciate an increase in your score!
Please note that we cannot update the paper after Wednesday, the 22nd of November, 2023.

Best regards,
Authors

---

> ### Comment · Reviewer_zSMw · 2023-11-22
>
> I appreciate the author's work in editing the paper in response to my comments, and for their openness to the suggestions from the reviewers. However, I do not plan to increase my score. In my opinion, one of the main tenets of the paper -- that sparse matrix sketching is useful in bounding the compressibility of large sparse matrices is incorrect. Even with the current edits, which significantly improve things, I feel that the paper should not even introduce this technique, as it does not add anything over trivial counting arguments. Introducing it, and highlighting it as a main technique, including in the title, muddles the contribution of the paper significantly. I encourage the authors to reformulate the paper and submit to another venue.
>
> Some more minor comments:
> - There is still an incorrect line in the paper that reads: "Counting the number of parameters in small dense matrices
> is more efficient in terms of parameters than counting the number of large sparse matrices, thus
> providing a way of evasion of the combinatorial explosion" -- I think the authors might have missed this when revising.
> - In the revision, I would recommend reworking the proof to apply beyond Gaussian distributions to the most general model possible (e.g. independent, mean 0 weights). Again, restricting to Gaussian muddles the contribution, as it implicitly implies that something about the Gaussian distribution itself is being used.

---

### Meta-Review · Area_Chair_5JJ1 · 2023-12-06

**Metareview:**

This paper explores the generalization bounds of pruned neural networks using random matrix theory and matrix sketching. The writing and presentation still require significant improvement, impeding reader understanding, with numerous typos persisting even after the revision.

While the reviewers acknowledge the novelty of using matrix sketching to derive generalization bounds, they raise concerns about the justification of the proposed method's motivation and the validity of the new bounds. The original claim that sparse matrix sketching can improve generalization bounds for pruned neural networks is deemed incorrect. Consequently, the motivation and the conclusion regarding using matrix sketching in pruning become questionable.

Reviewers also note a lack of detailed and quantitative comparisons of the derived bounds with existing generalization bounds. Some bounds appear vacuous, with excessively large constants and a failure to demonstrate improved dependence on the spectral norms of weight matrices.

Additionally, the authors are encouraged to apply their methods beyond Gaussian distributions, as claimed in the paper, to the most general models.

**Justification For Why Not Higher Score:**

The claimed contribution seems to be vacuous and the main motivation of the paper is questionable.

**Justification For Why Not Lower Score:**

N/A

---

### Decision · Program_Chairs · 2024-01-16

Reject